# Derailing the aspartate pathway of *Mycobacterium tuberculosis* to eradicate persistent infection

Erik J. Hasenoehrl[1], Dannah Rae Sajorda[1], Linda Berney-Meyer[1], Samantha Johnson[1], JoAnn M. Tufariello[1,6], Tobias Fuhrer [2], Gregory M. Cook[3,4], William R. Jacobs Jr.[1,5] & Michael Berney [1]

A major constraint for developing new anti-tuberculosis drugs is the limited number of validated targets that allow eradication of persistent infections. Here, we uncover a vulnerable component of *Mycobacterium tuberculosis (Mtb)* persistence metabolism, the aspartate pathway. Rapid death of threonine and homoserine auxotrophs points to a distinct susceptibility of *Mtb* to inhibition of this pathway. Combinatorial metabolomic and transcriptomic analysis reveals that inability to produce threonine leads to deregulation of aspartate kinase, causing flux imbalance and lysine and DAP accumulation. *Mtb's* adaptive response to this metabolic stress involves a relief valve-like mechanism combining lysine export and catabolism via aminoadipate. We present evidence that inhibition of the aspartate pathway at different branch-point enzymes leads to clearance of chronic infections. Together these findings demonstrate that the aspartate pathway in *Mtb* relies on a combination of metabolic control mechanisms, is required for persistence, and represents a target space for anti-tuberculosis drug development.

[1] Department of Microbiology and Immunology, Albert Einstein College of Medicine, Bronx, NY, USA. [2] Institute of Molecular Systems Biology, Swiss Federal Institute of Technology, Zurich, Switzerland. [3] Department of Microbiology and Immunology, School of Biomedical Sciences, University of Otago, Dunedin 9054, New Zealand. [4] Maurice Wilkins Centre for Molecular Biodiscovery, The University of Auckland, Private Bag 92019, Auckland 1042, New Zealand. [5] Department of Molecular Genetics, Albert Einstein College of Medicine, Bronx, NY, USA. [6] Present address: Center for Microbial Pathogenesis, Institute for Biomedical Sciences, Georgia State University, Atlanta, GA 30303, USA. Correspondence and requests for materials should be addressed to M.B. (email: michael.berney@einstein.yu.edu)

*M*ycobacterium tuberculosis (*Mtb*), the causative agent of tuberculosis (TB), is the deadliest pathogen worldwide, with 1.8 million deaths in 2016, and is the source of severe societal and economic burdens[1]. *Mycobacterium tuberculosis* is able to establish a devastating chronic infection in humans, where it can withstand host adaptive immunity and survive in granulomas, a phase commonly referred to as persistence. Many drugs are ineffective during persistence, most likely due to reduced penetrance into the devascularized granuloma environment[2] and enrichment of phenotypically drug-tolerant cells. As a result, treatment regimens require a lengthy 6-months of combination therapy to clear the infection, which greatly increases the risk of antibiotic resistance acquisition[1,3].

Recent efforts to target *Mtb* during persistence have focused on inhibiting previously ignored metabolic pathways[4,5]. Several studies have demonstrated that central carbon metabolism (CCM) is required during persistence, and several targets have been identified as essential in both acute and chronic infections in mice[4–11]. Importantly, biosynthetic pathways that utilize TCA cycle intermediates, such as amino acid biosynthesis, are also required for *Mtb* survival[12]. However, none of the amino acid biosynthetic pathways of *Mtb* have yet been shown to be essential during chronic infection and it is largely unknown if such building blocks can be scavenged from the host during persistence.

We recently reported that methionine biosynthesis in *Mtb* is required to establish infections in immunocompetent and immunocompromised mice[13]. Methionine, along with isoleucine, threonine, and lysine make up the family of essential amino acids synthesized from aspartate via the aspartate pathway. This pathway is also involved in the biosynthesis of the peptidoglycan building block, diaminopimelate (DAP), and the cofactor, S-adenosyl methionine (SAM); both of which are required for survival in mycobacteria[13,14]. Because of its branched nature and pivotal role in essential cellular processes (cell wall biosynthesis, translation, and one-carbon metabolism), metabolic regulation of the aspartate pathway is likely important to meet precursor requirements during growth and persistence in the host[15].

The apparent essentiality of the numerous metabolic products and absence of the aspartate pathway in humans and animals[15] led us to investigate the effects of inhibition of the aspartate pathway in *Mtb* and its requirement during persistence in the host. We hypothesized that disruption of the aspartate pathway leads to a metabolic imbalance and eventual collapse of this essential biosynthetic network. To investigate this, we employed a combinatorial approach of mycobacterial genetics, transcriptomics, metabolomics, biochemical analysis, and animal experiments. Here, we show that both homoserine auxotrophy (Δ*thrA*) and threonine auxotrophy (Δ*thrB*) are bactericidal, and that methionine and threonine are jointly required for survival and persistence. Transcriptomic and metabolomic studies revealed that threonine starvation leads to an unexpected accumulation of lysine, which the cell responds to by activating lysine export and a mycobacteria-specific degradation pathway. Finally, we demonstrate the absolute requirement of the branch-point enzymes homoserine dehydrogenase (ThrA, Rv1294) and homoserine transacetylase (MetX, formerly known as MetA, Rv3341) in both acute and chronic tuberculosis infections. Taken together our work identifies and characterizes a vulnerable component of *Mtb*'s persistence metabolism with its metabolic control mechanism and puts it on the map for drug discovery.

## Results

**Threonine and homoserine auxotrophy are bactericidal.** The aspartate pathway produces essential proteinogenic amino acids threonine, methionine, lysine, and isoleucine, the co-factor S-adenosyl methionine, and the cell wall component diaminopimelate. To dissect the individual contributions of pathway branches to metabolic homeostasis and pathogen survival, we attempted to delete selected branch-point enzymes to create auxotrophic strains in *Mtb*. These auxotrophic strains could then be used to study metabolic alterations in vitro and serve as metabolic probes in vivo. We have previously shown that MetX (formerly known as MetA), a homoserine transacetylase that converts homoserine to *O*-acetyl-L-homoserine, is essential for *Mtb* survival[13]. Upstream of MetX lies an important branch point of the aspartate pathway because homoserine is needed for both threonine and methionine biosynthesis (Fig. 1a). In order to dissect the effects of combined methionine/threonine and single threonine or methionine auxotrophy, we made null deletion mutants of homoserine dehydrogenase (ThrA) and homoserine kinase (ThrB) in *Mtb* H37Rv, leading to the mutant strains Δ*thrA* and Δ*thrB*, respectively. Deletion of *thrB* yielded a threonine auxotroph (Fig. 1b), which was killed rapidly in vitro in the absence of threonine supplementation (Fig. 1c). Death in this mutant could not be rescued by addition of isoleucine. Deletion of *thrA* was bactericidal and could be rescued by either the addition of homoserine or by a combination of methionine and threonine (Fig. 1d). Interestingly, the addition of either methionine or threonine alone in Δ*thrA* led to a statistically significant slowdown of cell killing (Fig. 1e).

**Threonine auxotrophy leads to accumulation of lysine and catabolites.** We next sought to elucidate the cellular consequences to inhibition at the threonine- homoserine-methionine branch-point. We employed a combinatorial transcriptomic and metabolomics approach. mRNA transcript levels of the Δ*thrB* and Δ*thrA* mutant strains were measured by RNA-seq and microarray analysis, respectively, in time-course experiments in growth media lacking the amino acids required for survival (Fig. 2). The Δ*thrB* and Δ*thrA* strains have an expression profile similar to that previously observed in methionine starvation (Supplementary Data 2)[13] and general nutrient starvation (Supplementary Data 2)[16], as evidenced by the downregulation of lipid biosynthesis and respiration-related genes, and upregulation of pyruvate and succinate metabolism (Fig. 2). However, there were also significant differences in gene expression that indicated killing by threonine and homoserine auxotrophy occurs via a mechanism distinct from that mediated by methionine or general starvation[13,16]. Threonine and homoserine auxotrophy each led to a particularly strong and early (within 2 days) response (upregulation) in redox stress gene expression (Fig. 2). The strongest changes were observed in genes involved in lysine degradation and export pathways. The third-most upregulated gene (8-fold) was the lysine-ε aminotransferase, *lat* (Rv3290c) (Fig. 2), which catalyzes degradation of lysine to α-aminoadipic acid semialdehyde (Fig. 1a). The *lat* gene has been identified as a persistence-related gene in *Mtb* and *M. smegmatis*[16–18]. The concurrent upregulation of *pcd* (Rv3293) (Fig. 2), which further degrades α-aminoadipic acid semialdehyde to α-aminoadipate, and Rv1986, a probable lysine exporter[19], suggests a general upregulation in lysine disposal pathways.

To investigate if these substantial transcriptional changes also translate into metabolic changes, we characterized the intracellular and extracellular metabolomes by ultra-performance liquid chromatography-mass spectrometry (UPLC-MS). In accordance with the upregulation of genes for lysine disposal, the Δ*thrA* mutant in media lacking homoserine displayed a significant increase (up to 100-fold) of all identifiable metabolites in the lysine biosynthesis pathway (Fig. 3a). Diaminopimelate, lysine, α-aminoadipate, and N-acetylysine were all found in at least 10-fold

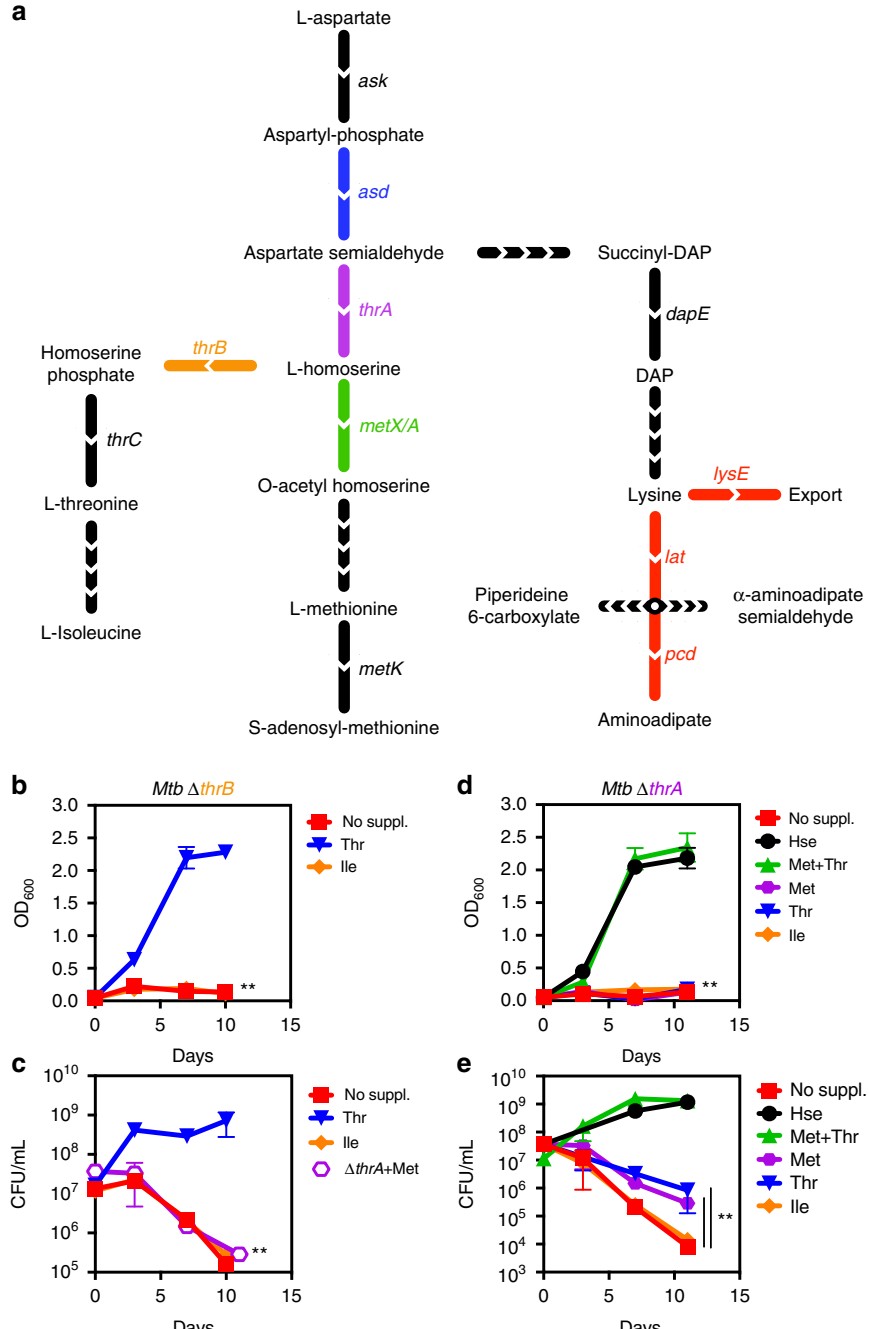

**Fig. 1** Inactivation of homoserine dehydrogenase and homoserine kinase is bactericidal. **a** The aspartate family amino acid biosynthesis pathway (aspartate pathway) in *M. tuberculosis* including the lysine degradation (aminoadipate) pathway. Ask: aspartate kinase, Asd: aspartate semialdehyde dehydrogenase, ThrA: threonine dehydrogenase, MetX: homoserine transacetylase, ThrB: homoserine kinase, ThrC: threonine synthase, MetK: S-adenosyl methionine synthase, DapE: Succinyl-diaminopimelate desuccinylase, LysE: lysine exporter, Pcd: aminoadipate semialdehyde dehydrogenase, Lat: lysine transacetylase. Two mutants were constructed, one threonine auxotroph *Mtb ΔthrB* and *Mtb ΔthrA, a* strain auxotrophic for both, threonine and methionine. **b, d** Growth measured by optical density OD$_{600}$. **c, e** Survival curves measured by colony forming units (CFU). **b** Deletion of *thrB* can be chemically complemented by threonine (Thr) (blue inverted triangles) but not isoleucine (Ile). **c** Deletion of *thrB* is bactericidal and mirrors the kill curve of *Mtb ΔthrA* supplemented with methionine (Met) (open rhombus). **d** Deletion of *thrA* can be chemically complemented by homoserine (Hse) (black circles) or a combination of threonine and methionine (blue triangles), but not by threonine, isoleucine (orange diamonds) and methionine alone (purple rhombus). **e** Deletion of *thrA* is bactericidal. Cell death of *Mtb ΔthrA* can be slowed by addition of threonine or methionine but not isoleucine, indicating that inhibition of each individual branch (threonine and methionine) contributes to killing. All values are the average of three biological replicates ($n = 3$) ± s.d. (error bars show standard deviation) and are representative of a minimum of two independent experiments. **p-value $< 0.01$ in Student's *t*-test. Source data are provided as a Source Data file

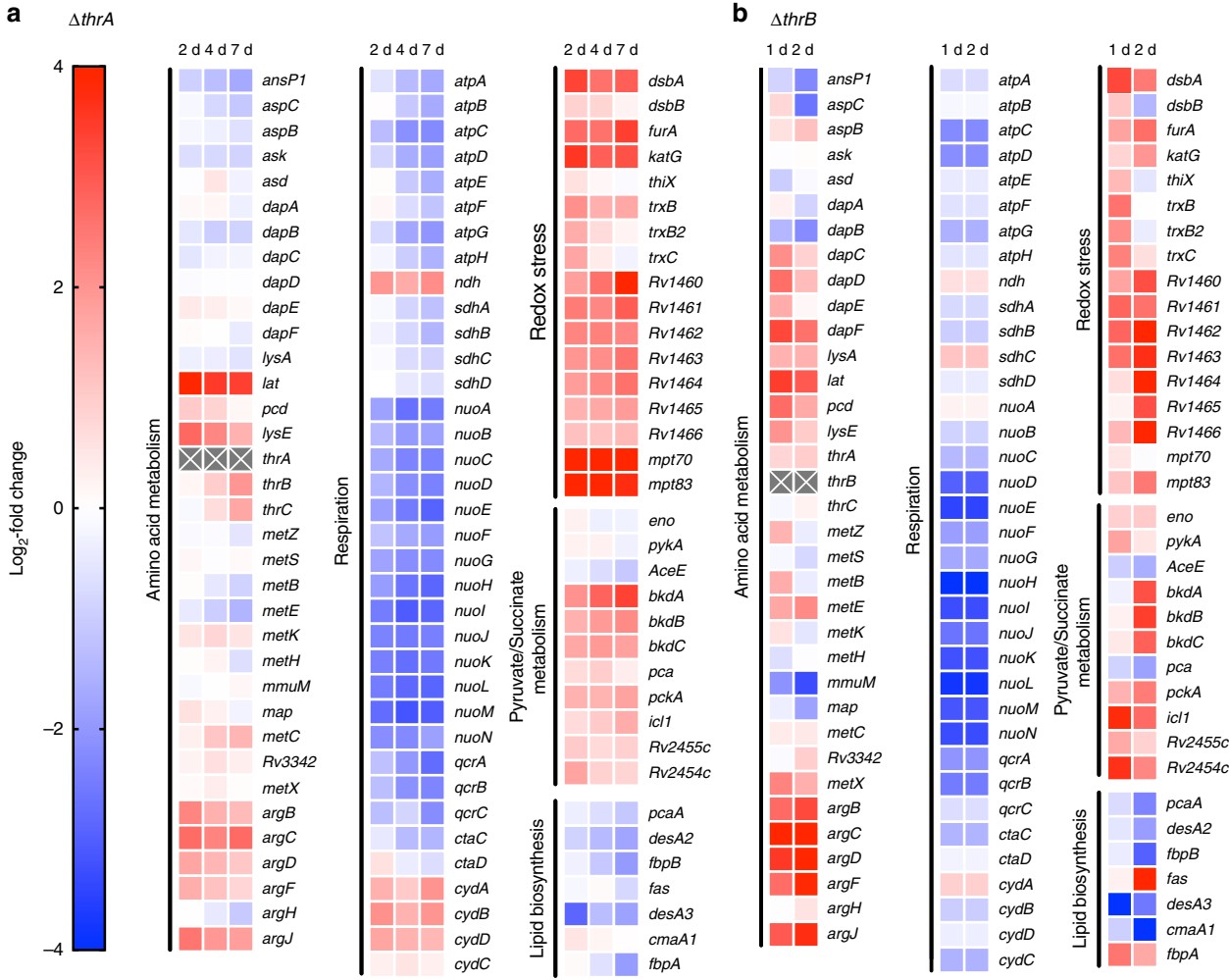

**Fig. 2** Time-course transcriptome of *Mtb* Δ*thrA* and Δ*thrB* mutants. Time-course transcriptomic profile of the Δ*thrA* (**a**) and Δ*thrB* (**b**) mutants during homoserine and threonine starvation, respectively. Samples from $n = 4$ (Δ*thrA*) and $n = 3$ (Δ*thrB*) biologically independent replicates were harvested on days 0, 2, 4, and 7 for Δ*thrA* and days 0, 1, and 2 for Δ*thrB*, and RNA was extracted. Scale is represented in log$_2$-fold change compared to day 0 in supplemented media. Genes in relevant pathways are presented. Gene expression ratios were considered statistically significant if there was a *p*-value less than 0.05. *p*-values for each gene and timepoint can be accessed via the GEO database (accession no. GSE119105, GSE119106, GSE119107)

greater abundance after 24 h of starvation of the Δ*thrA* mutant (Fig. 3a). Furthermore, lysine, but not other pathway intermediates, was found to accumulate extracellularly (Fig. 3b), suggesting increased activity of a lysine exporter. Over a period of 4 days the extracellular concentration of lysine increased to 160 μM after which it stayed constant (Supplementary Fig. 2). Analysis of the metabolomic response of threonine auxotrophy alone (*Mtb* Δ*thrB*) showed a similar pattern of intracellular metabolite abundances (Fig. 4), but in addition, *Mtb* Δ*thrB* also accumulated homoserine (Supplementary Fig. 3), the adduct of the *thrB* encoded threonine kinase (Fig. 1a), which nicely confirms the metabolic block at ThrB. In line with this homoserine accumulation, we observed increased expression of *metX* (Fig. 2b). Lysine export was also observed in the Δ*thrB* strain, albeit at a reduced rate compared to Δ*thrA* (Supplementary Fig. 4), which fits with the fact that carbon rerouting in Δ*thrB* occurs not only toward lysine but also homoserine. Homoserine was proposed to be toxic to *M. tuberculosis*[20], and we have confirmed its toxicity for the MTB Complex strain *M. bovis* BCG (Supplementary Fig. 5). We observed an adverse role of homoserine but not threonine on growth of *M. bovis* BCG at concentrations above 50 μg/ml (Supplementary Fig. 5), which is

in excellent agreement with the work published earlier[20]. These findings reveal accumulation of lysine, and homoserine in Δ*thrB*, that the cells attempt to manage by carbon-rerouting, lysine export and degradation. The lysine overload was particularly surprising given that most bacteria employ a lysine feedback loop that involves allosteric inhibition of aspartate kinase by lysine[21–27].

**Threonine is the main feedback regulator of *Mtb*'s aspartate pathway.** Next, we wanted to understand at a regulatory level, how inhibition of ThrA or ThrB could lead to increased lysine production. Aspartate kinase (AK) encoded by *ask* (Rv3709c) catalyzes the first committed step in the aspartate pathway, converting aspartate to aspartyl phosphate (Fig. 1). In *Escherichia coli*, there are three AK isoforms that are separately sensitive to feedback regulation by threonine, lysine/leucine, or methionine, respectively[21,23,24]. The actinomycete *Corynebacterium glutamicum* regulates the aspartate pathway via a single AK that is feedback inhibited by lysine and threonine, and transcriptional repression of homoserine dehydrogenase (*thrA*) by methionine[22,26]. In *M. tuberculosis*, purified AK has been shown

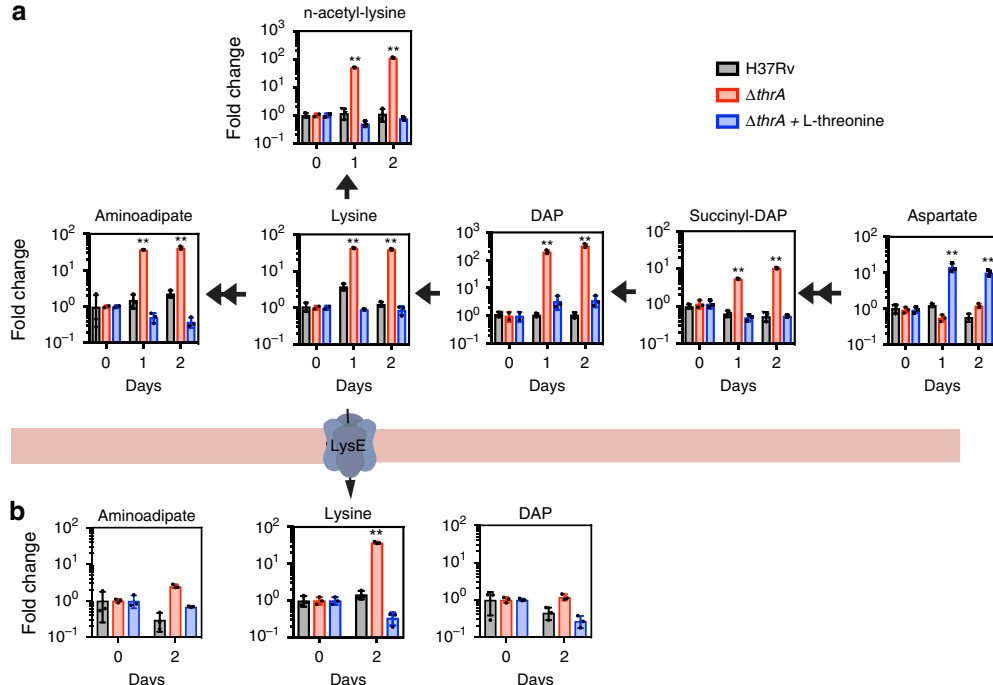

**Fig. 3** Endo and exometabolome of *Mtb ΔthrA*. **a** Starvation of the homoserine auxotroph *Mtb ΔthrA* in unsupplemented medium (red bars) leads to strong (up to 100-fold) intracellular accumulation of lysine and DAP and its biosynthetic intermediates, as well as downstream metabolites aminoadipate, and n-acetyl-lysine. Accumulation of the same intermediates is not observed in WT cells (*Mtb* H37Rv) under the same conditions (black bars). Addition of threonine to *Mtb ΔthrA* reverses accumulation of lysine and its intermediates (blue bars). Under the same conditions, aspartate but no other intermediate accumulates indicating that aspartate kinase is feedback inhibited by threonine. **b** Analysis of the exometabolome shows accumulation of lysine but no other intermediate in the supernatant of unsupplemented *Mtb ΔthrA* (red bars). Threonine addition stopped lysine export (blue bars). All values are the average of three biological replicates ($n = 3$) ±s.d. (error bars depict standard deviation) and are representative of a minimum of two independent experiments. **p-value < 0.01 in Student's t-test. Bars are overlaid with dot plot of individual values. Source data are provided as a Source Data file

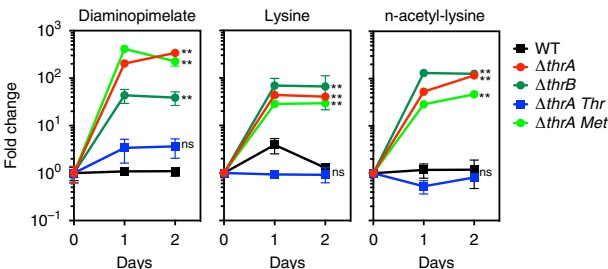

**Fig. 4** Threonine but not methionine deficiency leads to lysine overload. To confirm that threonine deficiency alone is responsible for accumulation of lysine and intermediates, the endo-metabolome of the threonine auxotroph *Mtb ΔthrB* was determined and compared to *Mtb ΔthrA*. Accumulation of DAP, lysine and n-acetyl-lysine was detected in both strains. Only threonine but not methionine reversed lysine accumulation in *Mtb ΔthrA*. All values are the average of three biological replicates ($n = 3$) ± s.d. (error bars depict standard deviation) and representative of two independent experiments. **p-value < 0.01 in Student's t-test

to be sensitive to threonine but not lysine[27], yet no other amino acids (e.g., methionine or homoserine) were tested nor was it investigated if these results translated to whole-cell physiology. Methionine and SAM biosynthesis in *Mtb* are likely regulated at the transcriptional level[13,28]. Consequently, addition of threonine should reverse lysine accumulation in *Mtb ΔthrA*. Indeed, threonine was capable of decreasing intracellular lysine accumulation, causing a concomitant accumulation of aspartate

(Fig. 3a). Furthermore, addition of methionine as a control, caused no change in lysine pathway metabolites (Fig. 4), indicating that the loss of threonine is the major driver for overproduction of lysine. This argues that L-threonine inhibition of AK is likely a primary metabolic control mechanism of the aspartate pathway.

**A lysine exporter responds to disruptions in lysine homeostasis**. We next investigated the unexpected export of lysine as an alternative mechanism to cope with lysine overload in the cell. Our transcriptomic analysis of the *ΔthrA* and *ΔthrB* mutants showed a strong upregulation of a putative lysine exporter *Rv1986* and arginine biosynthesis genes (Fig. 2a, b). Moreover, metabolomic data demonstrates that intracellular accumulation of Lys leads to export of both lysine and arginine (Arg) (Fig. 5a) but not other amino acids or metabolites (Supplementary Fig. 6). It has been shown in *C. glutamicum* that lysine accumulation is toxic, and that export of both Lys and Arg is carried out by the exporter LysE[29,30]. A bioinformatic analysis showed that *Mtb* encodes several proteins with homology to LysE of *C. glutamicum*, yet only Rv1986 is located adjacent to its canonical transcriptional regulator LysG (*Rv1985c*) (Fig. 5b)[19]. Hence, we hypothesized that Rv1986 serves the purpose of an overflow mechanism to prevent intracellular lysine overload. To test the role of LysE in amino acid export of *Mtb*, we measured the ability of *Mtb* to export toxic amino acid analogs in the presence and absence of the LysE exporter. This is a commonly used strategy to characterize amino acid exporters in whole cells[31]. We deleted the genes *lysE* (*Rv1986*) and *lysG* (*Rv1985c*) in *Mtb* H37Rv, resulting

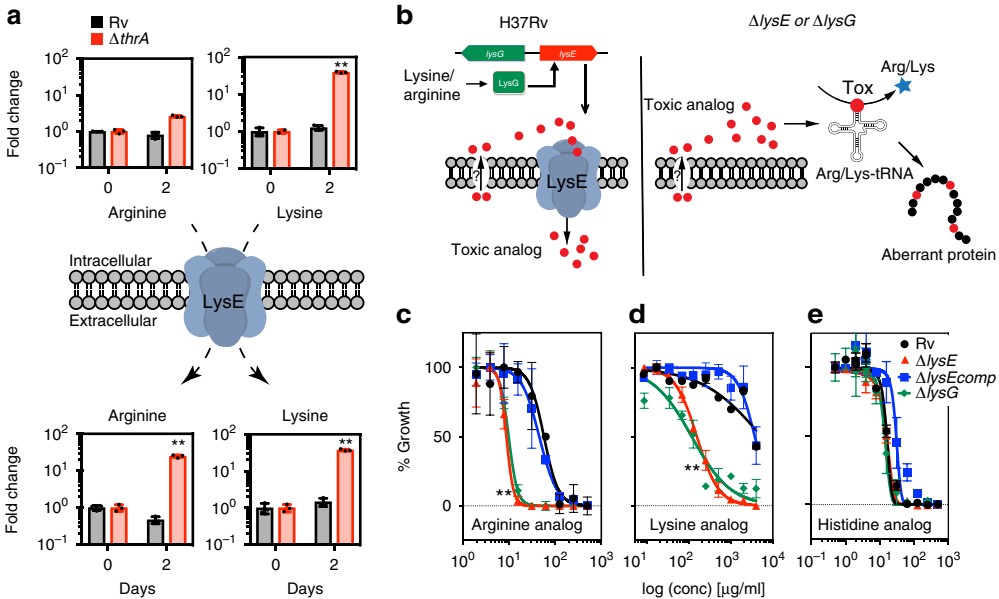

**Fig. 5** Lysine and arginine are exported via the lysine permease LysE. **a** Accumulation of arginine and lysine in the supernatant of *Mtb ΔthrA*, indicated that *Mtb* harbors a lysine/arginine permease LysE. Rv1986 was identified bioinformatically as a LysE homolog and Rv1985c as its regulator LysG[19]. Both genes were individually deleted from *Mtb* H37Rv to yield strains *ΔlysE* and *ΔlysG* (Supplementary Table 1) and *ΔlysE* was complemented with a plasmid containing lysE under its native promoter yielding strain *ΔlysEcomp*. **b** Illustration of how inactivation of LysE leads to hypersusceptibility to toxic analogs. **c, d, e** Exporter specificity for lysine and arginine was tested by MIC experiments on *Mtb* H37Rv (green), *Mtb ΔlysE* (blue), *Mtb ΔlysEcomp* (black) and *Mtb ΔlysG* (red) using toxic analogs of **c** arginine (canavanine), **d** lysine (2,6-diaminohex-4-ynoic acid and) and **e** histidine (ß-(2-Thiazolyl-DL-alanine). *Mtb ΔlysE and ΔlysG* are both hypersusceptible to toxic analogs of lysine and arginine but not histidine. All values are the average of three biological replicates (*n* = 3) ± s.d. (error bars depict standard deviation) and are representative of a minimum of two independent experiments. \*\**p*-value <0.01 in Student's *t*-test. Bars are overlaid with dot plot of individual values. Source data are provided as a Source Data file

in mutants *ΔlysE* and *ΔlysG*. Using these strains, we determined the minimum inhibitory concentration (MIC) of toxic amino acid analogs of lysine (2,6-diaminohex-4-ynoic acid), arginine (canavanine) and histidine (ß-2-Thiazolyl-DL-alanine) (Fig. 5c, d, e). These compounds (Supplementary Fig. 1) have previously been shown to be toxic analogs in other bacteria and act as substrates for their cognate amino acid exporters[32–34]. Toxicity arises from their aberrant incorporation into proteins during translation, and deletion of their analogs' exporters should increase susceptibility (Fig. 5b). Indeed, we found that deletion of both *lysE* and *lysG* resulted in a greater than 10-fold reduction in the MIC of canavanine and 2,6-diaminohex-4-ynoic acid (Fig. 5c, d), but there was no change in the MIC of the histidine analog control (Fig. 5e). The *lysE* phenotype was rescued when genetically complemented (Fig. 5c, d) or when the cognate amino acid (e.g., arginine) was added as a competitive substrate (Supplementary Fig. 7). The results demonstrate that LysE exports lysine and arginine as predicted by its homology and similarity (32% identity determined by Basic Local Alignment Search Tool (BLAST)) to the *C. glutamicum* LysE permease and that LysG is the activator of LysE under the conditions tested (Fig. 5). We also found evidence that lysine disposal is relevant in *Mtb* during growth in standard growth medium under non-auxotrophic conditions. During the transition from exponential to stationary phase, lysine and aminoadipate are elevated in wild-type *Mtb* H37Rv (Supplementary Fig. 8) and the *lysE* deletion mutant shows a significant growth retardation under these conditions (Supplementary Fig. 9).

Taken together, these data indicate that *Mtb* utilizes lysine export and degradation pathways as a metabolic relief valve to prevent disruption of aspartate pathway homeostasis and maintain essential precursor production. Such a metabolic mechanism to maintain precursor availability for growth and

survival is unprecedented in *M. tuberculosis*. It is tempting to speculate that prevention of lysine export, could synergize with inhibition of threonine production, and could be exploited to enhance bactericidal activity. To test this notion, we attempted to construct a double deletion strain *ΔthrAΔlysE*. However, multiple attempts were unsuccessful pointing to a synthetic lethal interaction between these two pathways.

**Aspartate pathway enzymes are required during chronic infections.** Given the potent bactericidal phenotypes observed in several auxotrophs in vitro, we next evaluated the need for aspartate family amino acid biosynthesis during host infection. We first tested the ability of the threonine/methionine auxotroph *Mtb ΔthrA* to establish an infection in immunocompetent (C57BL/6) and immunocompromised (SCID) mice via aerosol infection. Similar to *ΔmetX*[13], we found that *ΔthrA* mutant was incapable of establishing an infection in either C57BL/6 (Supplementary Fig. 10a, b) or SCID mice (Supplementary Fig. 10c, d). To test whether threonine auxotrophy alone prevents the establishment of infection, we also tested *Mtb ΔthrB* in C57BL/6 and found this strain to be avirulent too (Supplementary Fig. 11). However, as *Mtb* can maintain a persistent infection it is also necessary to test target vulnerability during an established, chronic infection[35]. As specific target inhibitors are not yet available, a genetic approach was employed. We constructed conditional knockdowns (cKD) of ThrA (*ΔthrA*-DUC) and MetX (*ΔmetX*-DUC), using the tetracycline-induced Dual-Control (DUC) system[35]. Both constructs are transcriptionally regulated by an anhydrotetracycline (atc) -inducible transcriptional repressor (T38), and are simultaneously repressed by atc-induced proteolytic degradation (DUC)[36,37]. The knockdown efficiency was validated in vitro by measuring the effect of increasing

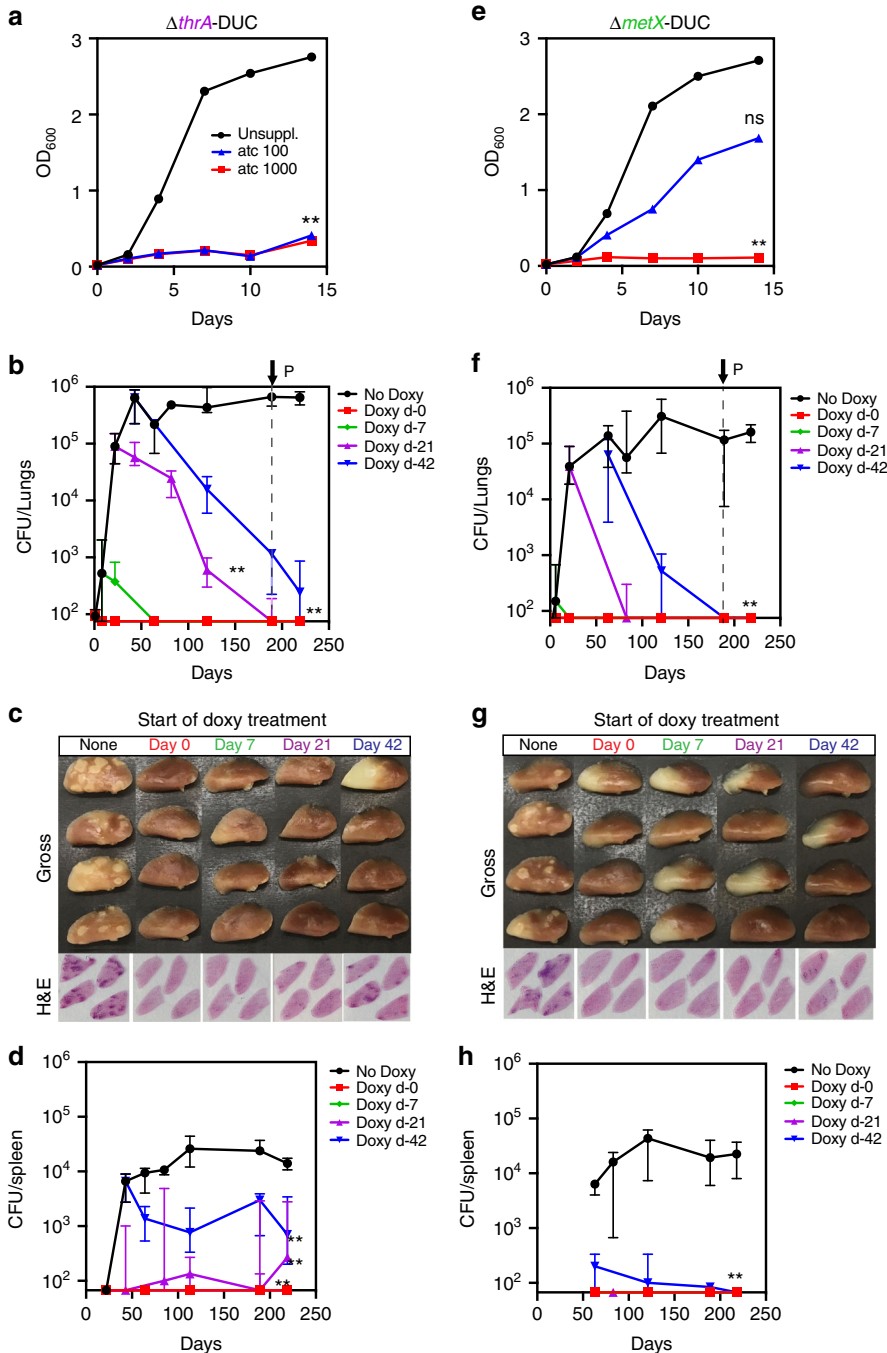

**Fig. 6** Aspartate pathway enzymes are essential during acute and chronic mouse infections. Conditional knockdowns were constructed of two enzymes of the aspartate pathway: homoserine dehydrogenase (ΔthrA-DUC) (**a–d**) and homoserine transacetylase (ΔmetX-DUC) (**e–f**). **a, e** Effect of gene knockdown/protein degradation on growth was tested in vitro in unsupplemented 7H9 OADC with different concentrations of anhydrotetracycline (atc). **b, f** Mice were aerosol infected with either ΔthrA-DUC3 or Δmet-DUC5 to deliver ~100 CFU/lungs. Groups of mice were switched to doxycycline containing chow (2000 ppm) at day 0 (red square), day 7 (green diamond), day 21 (purple triangle) and day 42 (blue inverted triangle) to induce knockdown. One group (black) was left untreated throughout the whole experiment. At given time points, 4 mice per group were sacrificed and bacterial burden of lungs (**b, f**) and spleens (**d, h**) determined by CFU counts of homogenized, serially diluted tissue. Gross lung pathology and Hematoxylin and Eosin (H&E) staining of lungs at day 189 are shown under (**c, g**). X-axis crosses Y-axis at limit of detection. Error bars show standard deviations from n = 4 mice. **p-value < 0.01, *p-value < 0.05 in Student's t-test. Source data are provided as a Source Data file

concentrations of atc on growth of the recombinant strains in culture broth media (Fig. 6a, e). Growth of the DUC strains in medium without atc was indistinguishable from the parental strain, demonstrating genetic complementation to WT levels (Supplementary Fig. 12a). While the parental H37Rv strain showed no sensitivity to atc (Supplementary Fig. 12b), growth

was inhibited in both knockdown strains (Fig. 6a, b). Next we tested the vulnerability of *Mtb* to inhibition of these enzymes during different stages of infection in C57Bl/6 mice. Cohorts of each infection group were started on doxycycline at 0, 1, 3, or 6 weeks after infection, which allowed us to study the in vivo essentiality of these enzymes during both the acute and chronic

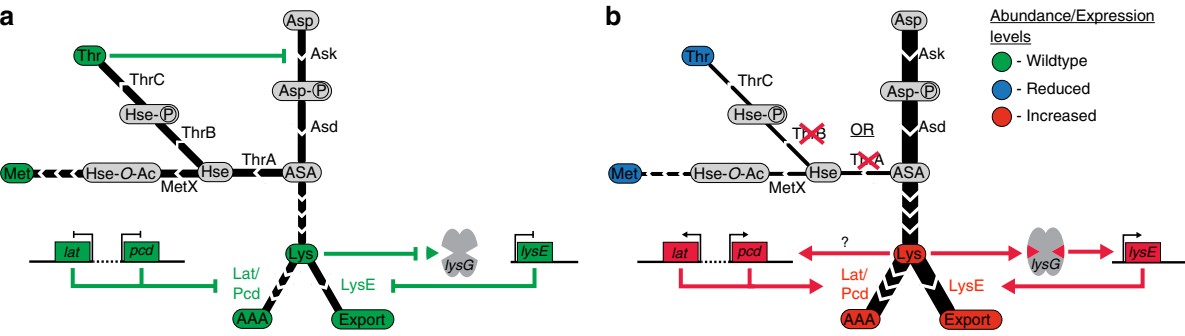

**Fig. 7** Schematic of aspartate pathway regulation. **a** Wildtype levels of threonine lead to proper feedback regulation of Ask, and subsequent wildtype levels of lysine and its degradation (Lat/Pcd) and export (LysE) pathway enzymes and metabolites. **b** Disruption of threonine biosynthesis causes derepression of Ask leading to an accumulation of Lysine, which in turn activates the lysine degradation and export pathways. Nomenclature: Metabolites in rounded boxes: Asp: aspartate, Asp-P: aspartate-phosphate, ASA: aspartate-semialdehyde, Hse: Homoserine, Hse-P: homoserine-phosphate, Thr: Threonine, HsE-O-Ac: O-acetyl-homoserine, Met: methionine, Lys: lysine, AAA: aminoadipate, Enzymes: Ask: aspartate kinase, Asd: aspartate semidaldehyde dehydrogenase, ThrA: homoserine dehydrogenase, ThrB: threonine kinase, ThrC: threonine synthase, MetX: homoserine O-acetyl-transferase, LysE: lysine exporter, Lat: lysine aminotransferase, Pcd: aminoadipate-semialdehyde dehydrogenase. Genes in square boxes. Color coding: green: wildtype levels, blue: reduces levels, red: increased levels

stages of infection (Fig. 6b, f). For both cKD strains, administration of doxycycline immediately after infection resulted in a total bacterial clearance (below detection limit) within one week after treatment initiation. Initiation of doxycycline treatment at the later time points resulted in clearance (below detection limit) of the infection from both lungs and spleens. By the end of the experiments, bacilli could no longer be cultured in the ΔmetX-DUC (Fig. 6f) infected mice, and only ΔthrA-DUC infected mice that began doxycycline treatment at week 6 still showed detectable levels of bacteria (Fig. 6b). The total bacterial burden had been reduced by 4-logs in the lungs and >1-log in the spleens (Fig. 6d, h, respectively). Inspection of gross pathology and H&E staining of lungs harvested 189 days after infection (indicated by arrow P and dotted line in Fig. 6b, f) reveal typical tuberculous lesions in untreated mice with high bacterial burden and absence of lesions and inflammatory foci in mice that were switched to doxy chow (Fig. 6c, g). These experiments demonstrate the essentiality of the aspartate pathway in both acute and chronic Mtb infections in animals and show that Mtb is unable to scavenge nutrients from the host to complement these auxotrophies in vivo.

## Discussion

Significant efforts to identify new drug targets in Mtb have focused on the metabolism of essential nutrients. Mtb is an amino acid prototroph that has developed nominal machinery to obtain nutrients from the host, unlike many other intracellular parasites[12,38] and consequently, Mtb appears to be particularly susceptible to inhibition of amino acid biosynthesis[12]. Moreover, the biosynthetic enzymes for essential amino acids are largely absent in humans and mammals, making amino acid biosynthetic pathways in Mtb attractive drug targets[12,39–42].

To date, in vitro transposon sequencing (Tn-seq) screens have been used to predict gene essentiality in Mtb[43,44] and have guided target selection in drug discovery. The usefulness of Tn-seq screens for the TB research field is undisputed, yet their limitations for drug discovery need consideration. A recent study highlighted the impact of growth medium on inferred gene essentiality[45] and illustrated how in vitro Tn-seq screens can be limited predictors of in vivo essentiality and cannot differentiate between lethality, bacteriostasis, slow growth phenotypes or conditional essentiality, all information that is crucial for assessing druggability. Since M. tuberculosis is an intracellular

pathogen, essentiality and druggability of targets have to be tested during chronic infection in order to prevent false targets being taken forward for drug development.

We here focused on the aspartate pathway, because it produces multiple essential building blocks for vital cellular functions in Mtb, such as cell wall biosynthesis, translation, and one-carbon metabolism. The rapid cell death phenotypes observed for auxotrophs of threonine (this study), homoserine (this study) and methionine[13], points to a distinct susceptibility of Mtb to inhibition of this pathway. The mechanism of action for the lethality of threonine and homoserine auxotrophy appear to be multilayered. Cell death kinetics due to the loss of proteinogenic amino acids is likely accelerated by the accumulation of intermediates of lysine biosynthesis and homoserine (only in thrB mutant).

By studying the metabolic response to aspartate pathway inhibition, we discovered that Mtb employs a combination of feedback control, overflow metabolism and catabolic action to prevent flux imbalances and ensure balanced precursor production. We show here corroborating evidence that the committed step of the aspartate pathway in Mtb, aspartate kinase, is controlled by allosteric feedback of threonine, but not by lysine or methionine. As Mtb lacks the ubiquitously used lysine-AK feedback loop, it must employ other regulatory mechanism to control production of proteinogenic lysine and the cell-wall building block diaminopimelate. By studying the adaptive metabolism of homoserine and threonine auxotrophic strains, we discovered two compensatory mechanisms that help Mtb to control the flux through this pathway branch: lysine degradation and export (Fig. 7b).

The lysine catabolic pathway catalyzed by Lat and Pcd is a unique feature of mycobacteria (Fig. 7). In a survey of 1362 bacteria, the aminoadipate branch was found to exist in only 36 microorganisms, predominately species of mycobacteria. This catabolic pathway seems to be a remnant of the cephamycin C biosynthetic pathway found in Streptomyces[46], yet mycobacteria have only retained the first two reactions (mycobacteria do not produce cephamycin C). Another unique feature of this pathway in mycobacteria is the localization of lat in an operon with the alternative sigma factor sigF (Rv3286c) and lrp (Rv3291c)[47] the putative transcriptional regulator of lat. SigF is important under various stress conditions, including osmotic stress, pH, heat-shock, oxidative stress[48,49], and during infection[50,51]. Expression of lrp and lat genes is enriched in a persistence model[17], as part of the stringent response[16,18] and whenever Mtb transitions from replicating to non-replicating conditions (e.g., starvation or drug

inhibition)[16,17]. These observations fit well with the increased abundance of lysine and aminoadipate observed in stationary phase cells (Supplementary Fig. 8). It is tempting to speculate that *Mtb* has lost lysine feedback control at the AK level in order to allow increased lysine flux to act as a signal for cellular stress. In this model, elevated intracellular lysine levels would cause upregulation of the Lat degradation pathway, which generates aminoadipate and leads to the activation of the stringent response and persistence. Alternatively, lysine catabolism and its potential toxic intermediate(s) may serve as a means to control growth under conditions where replication could be lethal, similar to the role of Toxin-Antitoxin modules[52] or methylglyoxal metabolism in persister cell formation[53]. Potential candidates for toxic intermediates are aminoadipate semialdehyde and piperideine-6-carboxylate (Fig. 1a), as both have been shown to be cytotoxic[54,55].

The second mechanism that mycobacteria employ to compensate for a missing lysine-AK feedback loop is catalyzed by a dedicated lysine/arginine permease LysE, encoded by Rv1986. We show that LysE in *Mtb* is used as a metabolic relief valve to clear excess cytoplasmic lysine, as a result of threonine auxotrophy. This exporter also has a role under non-auxotrophic conditions as evidenced by the retarded exit from lag phase and slow growth phenotype of the *Mtb* Δ*lysE* mutant. Current literature shows that Rv1986 is induced during infection of resting and activated macrophages[56], as well as in a lysosomal exposure model[57] and it plays a role during latent TB infection in humans[58]. The exporter is a dominant target of IL-2 secreting memory T cells and is proposed to contribute to protective immunity in humans[58]. In fact, amino acids, specifically arginine, were shown to modulate human immunometabolism and T-cell responses[59]. Together, these data suggest that LysE-mediated export of arginine, and possibly lysine, might play a role in shaping *Mtb's* niche during infection.

Our current knowledge of the nutritional composition of the *Mtb* microenvironment in the host is still incomplete. How that environment evolves over the course of an infection is even less understood[6,13,35]. Auxotrophic bacterial strains are essential tools for identifying metabolic drug targets required during infection and probing the host environment for specific metabolites. However, because most *Mtb* auxotrophs fail to establish infection, their requirement for late-stage persistence cannot be investigated. To date, no data is available on the vulnerability of amino acid biosynthetic pathways during *Mtb* persistence in vivo. By utilizing conditional knockdowns of two branch-point enzymes (*metX* and *thrA*), we uncovered the aspartate pathway as being broadly required for persistent infection in mice. These data also clearly demonstrate the inability of *Mtb* to scavenge sufficient methionine, homoserine or threonine from the host throughout the course of infection. The observations presented here lay an important foundation for future drug development. We propose that the numerous essential enzymes identified in this pathway and their absence in eukaryotic cells make it particularly amenable to drug discovery efforts.

## Methods
**Mycobacterial strain and growth conditions**. All strains and plasmids used in this study are listed in Supplementary Tables 1. Mycobacterial strains were grown in Middlebrook 7H9 medium (Difco) supplemented with 10% (vol/vol) OADC enrichment (0.5 g oleic acid, 50 g albumin, 20 g dextrose, 0.04 g catalase, 8.5 g sodium chloride in 1 L water), 0.2% (vol/vol) glycerol, and 0.05% (vol/vol) tylaxopol (Sigma). Antibiotic selection media contained 75 μg/mL hygromycin B, 20 μg/mL Kanamycin, and/or 25 μg/mL streptomycin, as required. Supplemented media for culturing auxotrophic mutants contained 50 μg/mL methionine, 500 μg/mL threonine, and/or 50 μg/mL homoserine. For time-course starvation and supplement growth curve experiments, strains were grown in the presence of the required supplements to an $OD_{600}$ of 0.5, washed three times with PBS + 0.05% tylaxopol, and diluted into supplement free medium to an $OD_{600}$ of 0.01 (growth/kill curves) or 0.2 (metabolomics and transcriptomics). Samples from three independent

biological replicates were harvested on days 0, 2, 4 and 7 (Δ*thrA*) or days 0, 1, 2 (Δ*thrB*) for RNA extractions, and days 0, 1 and 2 for metabolite extractions (see below). Chronologically we studied the Δ*thrA* strain first and noticed that the main transcriptomic and metabolic changes are happening in the first 2 days. Hence, we focused on this timeframe for the rest of the study. Anhydrotetracycline (Acros) was dissolved in DMSO and used at the specified concentrations.

**Gene Knockout and complementation**. The genes *thrA* (Rv1294), *thrB* (Rv1296), *lysE* (Rv1986), and *lysG* (Rv1985c) were deleted in *Mtb* H37Rv by specialized transduction[60]. The *Mtb* H37Rv Δ*metX* strain (previously known as Δ*metA*) was constructed previously[13]. Transductants were recovered on selective medium containing hygromycin (75 μg/mL) and appropriate amounts of supplements when needed: homoserine (50 μg/mL) for *Mtb* Δ*thrA* and threonine (500 μg/mL) for *Mtb* Δ*thrB*. Mutations were confirmed by three-primer PCR using respective L, R, and Universal_uptag primers, listed in Supplementary Table 2. The H37Rv Δ*metX* and Δ*thrA* strains were complemented using the tetracycline-inducible dual-control (DUC) knockdown plasmids[61]. Briefly, each strain was complemented with pGMCK-q19-T38S38-P750-*yfg*-DAS+4 (pGMCK-*yfg*). The gene sequences were PCR-amplified using the respective P1 and P2 primers (Supplementary Table 2) and were recombined into pDO12A using BP clonase (Invitrogen), to generate the entry vectors pEN12A-*yfg*-DAS+4. The final construct was generated by 3-way gateway cloning with pEN23A-*yfg*-DAS + 4, pEN41A-T38S38, pEN12A-P750, and pDE43-MCK using LR clonase II (Invitrogen), to generate the final construct. This was transformed into the respective mutant strain to generate DAS complements whose expression of the complemented genes was under a TetOFF promoter. The H37Rv *metX*-DAS and *thrA*-DAS strains were further transformed with pGMCtKq28-TSC10M1-sspB to introduce Tet-inducible degradation of DAS+4 tagged proteins. Shine-delgarno sequences were mutated according to Supplementary Table 3, using Phusion Site-directed Mutagenesis kits (NEB). The H37Rv Δ*lysE* strain was complemented by transformation with pMV361 harboring a copy of Rv1986 (Supplementary Table 2). This DNA fragment was PCR amplified using primers Rv1986_fw_EcoR1 and Rv1986_Re_HindIII (Supplementary Table 2) and was cloned into pMV361 using EcoRI and HindIII restriction sites, resulting in plasmid pMV361-LysE.

**Metabolite extractions**. Samples of bacterial cultures in biological triplicates were harvested at given time points[13,62]. For intracellular metabolites, an equivalent of 10 mL culture at an $OD_{600}$ of 0.2 was rapidly filtered on a 0.22 μm nitrocellulose filter (Millipore). The filter papers were immediately placed in 2-mL screw cap tubes of 1 mL extraction solvent containing 20%/40%/40% (vol/vol) water/acetonitrile/methanol (Fisher) with approximately 500 μL of silica beads at −20 °C. The samples were homogenized twice in a Precellys Evolution (Bertin) at 4,500 rpm for 45 seconds, cooled to 5 °C using a Cryolys (Bertin), and with 5 min rest between cycles. Samples were centrifuged, and 750 μL of the extract was filtered through 0.22-μm Spin-X centrifuge filters (Corning, Life Technologies) at 10,621 g for 3 min, and stored at −80 °C. For extracellular metabolites, 1 mL of culture was collected and centrifuged, and 750 μL of supernatant was filtered twice through 0.22-μm Spin-X columns. Detergent in the samples were removed using Pierce detergent removal spin columns (Thermo), according to manufacturer's instructions. Metabolites were extracted by adding 200 μL sample to 800 μL of 50%/50% (vol/vol) acetonitrile/methanol (Fisher), the samples were spun down, as before, and the supernatant collected and stored at −80 °C until analysis.

**Metabolomics**. Metabolomics analysis was performed using an Acquity UPLC system coupled with a Synapt G2 quadrupole–time of flight hybrid mass spectrometer (Waters, Massachusetts, USA)[13]. Column eluents were delivered via electrospray ionization. UPLC was performed in a hydrophilic interaction liquid chromatography-mode (HILIC) gradient elution using an Acquity 1.7-μm amide column (2.1 × 100 mm)[63]. The flow rate is 0.5 mL/min with mobile phase A (100% acetonitrile) and mobile phase B (100% water), both containing 0.1% formic acid. In both positive and negative mode, the gradient began with 1% B until 1 min, ramped to 35% B by 14 min, then 60% B by 17 min, held at 60% B for 1 min, then ramped to 1% B by 19 min and held at 1% B to the end of run at 20 min. The mass spectrometer was operated in V mode for high sensitivity using a capillary voltage of 2 kV and a cone voltage of 17 V. The desolvation gas flow rate was 500 L/h, and the source and desolvation gas temperature were 120 °C and 325 °C, respectively. MS spectra were acquired in centroid mode from *m/z* 50–1,200 with a scan time of 0.5 s. Leucine enkephalin (2 ng/μL) was used as lock mass (*m/z* 556.2771 and 554.2615 in positive and negative experiments, respectively). Analysis was performed using MarkerLynx (Waters) with extended statistic function to identify statistically significant differences of metabolite abundances. A TargetLynx (Waters) database was compiled by running standards of metabolites of interest to determine retention times and m/z ratios. Calibration curves of compound mixes are run routinely to determine the linear range of metabolite detection. All samples were analyzed using TargetLynx with manual curation of peak areas where necessary. With abundances we refer to are raw intensity values for the area under the curve for each metabolite peak. Peak areas were used to calculate fold changes of metabolites relative to day 0. Metabolite abundances, retention times and m/z values can be found in Supplementary Data 1. Threonine and homoserine have the

same molecular mass. The retention time of homoserine is 0.1–0.2 min slower than threonine, however differentiation of the two peaks is difficult. Fragmentation pattern of the two compounds in positive mode is nearly identical while in negative mode there is one distinction (m/z 72.04). We have detected high amounts of $m/z = 120.0655$ in the ThrB strain but not in Rv or ThrA. Since ThrB is downstream of homoserine, we hypothesize that this mass must represent homoserine, because the bacterium does not grow without the addition of high concentration threonine. In the ThrA strain, as expected from a strain that cannot produce homoserine nor threonine, the m/z 120.0655 was undetectable.

**RNA extraction**. Samples of bacterial cultures were harvested in triplicate at appropriate times. 10 mL of culture was pelleted and the supernatant decanted. The cell pellet was resuspended in 1 mL of TRIzol (Invitrogen) and incubated overnight at 4 °C, before storage at −80 °C. The suspension was transferred to Fast-Prep Blue Cap tubes and processed twice for 45 s at speed 6 in a Fast-Prep apparatus (MP Biomedicals). After a brief incubation on ice, the debris was spun down, and the supernatant (750 μL) was processed for purification (described below).

**Microarray (Mtb ΔthrA)**. Total RNA was purified from the cell lysate using Direct-zol RNA MiniPrep (Zymo Research) with on-column DNase I according to the manufacturer's protocol. Total RNA (2 μg) was used to create cDNA labeled with aminoallyl dUTP (Sigma). Fluorescent Cy3 and Cy5 dyes (GE Healthcare) then were covalently attached to the aminoallyl tags. Each pair of differentially labeled probes was resuspended in 60 μL of hybridization buffer (500 μL formamide, 250 μL 20 × SSC, 5 μL 10% (wt/vol) SDS, 245 μL ultrapure water) and hybridized to the microarray slide overnight in a 42 °C water bath. Slides then were washed in increasingly stringent wash conditions (three times for 5 min each washing in 1 × SSC 0.1% SDS; three times 5 min each washing in 0.1 × SSC 0.1% SDS; three times for 5 min each washing in 0.1 × SSC, and a final dip in 0.22-μm–filtered Milli-Q water [Millipore]). Arrays were scanned in a Genepix 4000A scanner, and spots were quantified with TIGR Spotfinder. The data then were processed in TIGR MIDAS. Two-color tiff images of the microarrays were inspected visually for defects and obvious spatial biases. Before normalization, pin-tip (block) intensity box plots were used to detect slides exhibiting spatial and/or pin biases; after normalization, intensity box plots and MA plots were used to assess the effectiveness of normalization in correcting these biases. Radio-intensity plots were used to identify outlier slides and to assess the effectiveness of the normalization process, and Z-score histograms were used to look for slides with abnormal intensity distributions. Slides that did not pass the quality-control process were rejected, and repeat hybridizations were performed. After normalization (total intensity and LOWESS) the in-slide replicate spots were averaged before expression ratios were calculated. The results from four independent biological replicates including two dye swaps then were subjected to a t-test without false discovery correction in TIGR MeV software. The analysis was used as a ranking method. For a general overview, genes with expression ratios >2 and <0.5 and a P-value <0.05 were used for data interpretation.

**RNA-seq (Mtb ΔthrB)**. RNA from the clarified cell lysate samples were purified by Trizol®-chloforom precipitation according to the manufacturer instructions (Invitrogen). After purification remaining DNA was removed from the samples using the DNA-Free Turbo DNase Kit (Ambion) and RNA integrity was checked on a Bioanalyzer RNA 6000 Pico chip (Agilent). The ribosomal RNA (rRNA) was removed using the Bacterial RiboZero kit (Illumina), and removal was confirmed by Bioanaylzer as above. RNA-seq library preparation was conducted with the TrueSeq Stranded mRNA Library Prep (Illumina) according to manufacturer's instructions and the library was sequenced on a NextSeq 500 (Illumina) using the NextSeq 500/550 HO V2 (75 cycles) (Illumina). Illumina reads were mapped to the M. tuberculosis reference genome from NCBI (NC_000962.3) in the form of a SAM file using Bowtie2 (http://bowtie-bio.sourceforge.net/index.shtml), the SAM files were converted to coordinate-sorted BAM files using samtools, and reads over genes were counted using featureCounts[64]. Differential expression analysis was then conducted using R and the DESeq2 package.

**Mouse experiments**. Mouse studies were performed in accordance with National Institutes of Health guidelines following the recommendations in the Guide for the Care and Use of Laboratory Animals[65]. The protocols used in this study were approved by the Institutional Animal Care and Use Committee of Albert Einstein College of Medicine (Protocols #20120114 and #20180612).

Female SCID mice and female C57BL/6 mice (Jackson Laboratories) were infected via the aerosol route using a $1 \times 10^7$ cfu/mL mycobacterial suspension in PBS containing 0.05% tylaxopol and 0.004% antifoam. Infection yielded approximately 100 bacilli per lung as determined by quantification of lung bacterial loads at 24 h post-infection (four mice per group). Subsequently, four mice from each group were killed at days 1, 7, 21, 63, 84, 98 (C57BL/6) or 1, 14, 22, and 42 (SCID) to determine the bacterial burden in the lung and spleen. Six SCID mice per group were kept for survival experiments. For conditional knockdown experiments, groups of mice infected with each strain were started on 2000 ppm doxycycline-treated chow (TestDiet, Missouri, USA) at 0, 1, 3, and 6 weeks. Four mice from each group were sacrificed at given times, the lungs and spleens were harvested,

homogenized and plated on 7H10 media containing the appropriate supplement to determine the bacterial burden. All mice infected with Mtb were maintained under appropriate conditions in an animal biosafety level 3 laboratory.

**Pathology**. Lung samples were fixed in 10% neutral buffered formalin for 48–72 h and the subjected to paraffin embedment. Tissues were sectioned at 5 μm and stained with Hematoxylin and Eosin (H&E).

**MIC determination of toxic metabolite analogs**. Cultures of Mtb H37Rv, ΔlysE, ΔlysEcomp, and ΔlysG were grown to log-phase in unsupplemented media. Each toxic analog (Supplementary Fig. 1) was two-fold serially diluted in a sterile 96 well plate. In triplicate, each strain was diluted to an OD₆₀₀ of 0.001 in wells containing each of the drugs to a final volume of 0.2 mL. The plates were incubated at 37 °C for 7 days and the growth was measured by optical density in an Epoch plate reader (Biotek, Vermont, USA). Percent growth was calculated based on the density in the untreated wells. For arginine rescue experiments, arginine was added to each well at a concentration of 50 μg/ml.

**Reporting summary**. Further information on research design is available in the Nature Research Reporting Summary linked to this article.

## Data availability

All microarray and RNAseq raw data were deposited in Gene Expression Omnibus (GEO) (GSE119105 (ΔthrA), GSE119106 (ΔthrB), GSE119107 (SuperSeries)). Data underlying Figs. 1–6 are provided as Source Data files. All other data are available from the corresponding author upon reasonable requests.

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

## Acknowledgements

This study was financially supported by NIH funding AI119573 and AI139465 for M.B., D.R.S., and E.H. and T32-GM007288 and AI138483 for E.H. S.J. was supported by NIH grant T32 AI007501. G.C. was supported by Marsden Fund NZ. J.M.T. was supported by effect:hope (Leprosy Mission Canada) on behalf of the Research to Stop Neglected Tropical Disease Transmission (R2STOP) Initiative. W.R.J. was supported by AI026170. We thank Dirk Schnappinger for providing the plasmids for making the DUC constructs and Sabine Ehrt for helping out with doxy chow. We thank John Kim, Mei Chen, Annie Zhi Dai, Bing Chen, Robert Dubin, Xusheng Zhang, and Fabien Delahaye for technical support and Kevin Pethe for critical reading of the manuscript.

## Author contributions

E.J.H., G.M.C., W.R.J. and M.B. designed the research. E.J.H., D.R.S., L.B.M., S.J., J.M.T., T.F. and M.B. performed research. E.J.H. and M.B. analyzed data. E.J.H., G.M.C. and M.B. wrote the paper.

**Additional information**

**Competing interests:** The authors declare no competing interests.

