## [Peer Review File · Nature Communications]

Reviewers' comments:

Reviewer #1 (Remarks to the Author):

This manuscript describes work towards the characterization of threonine/isoleucine/methionine auxotrophs in *Mycobacterium tuberculosis*. Unfortunately, by technical and scientific reasons explained below the reviewers feel that this manuscript needs significant work to pass peer-review in *Nature Communications* or other high profile journals. Mainly, there is very little novelty in the results presented.

Technical issues-

All mass spec data is reported as: abundance, normalized abundance and fold-change. While this style of presenting data is very popular with microarrays, it has no direct use in metabolism. The authors must be aware that different compounds ionize differently and that needs to be determined with standard curves. Through the manuscript the authors refer to abundance, relative abundance and so on, without really communicating with clarity what are they referring to.

Another major issue that precludes acceptance of this study and the claims made by the authors is the lack of independent experiments. All data presented appears to have been generated once, with biological replicates. The reviewers have no trust that the results are in fact reproducible. The authors must show that their data are reproducible by carrying truly independent experiments (different day, different cells, different media, etc). This is particularly important for metabolic experiments as well as microbiological experiments.

Other Major Issues:

Page 12/13. The authors have confirmed that threonine and homoserine auxotrophy is lethal to *Mtb*. Although, these exact experiments have not been carried out before, *thrA*, *thrB* are essential in *Mtb* (Sasseti CM et al. 2003; Griffin JE et al. 2011; DeJesus MA et al. 2017). These results are expected based on the vast data on transposon mutants and the current understanding of amino acid biosynthesis in *Mtb*. In addition, the authors have confirmed that Ile does not rescue Thr auxotrophy. This is also expected based on the number of irreversible steps after Thr.

Page 13/14. As the authors did not express the metabolomics data as concentration (molar or g/mL) the authors cannot really conclude anything with regards to accumulation. 10-fold change in lysine ion counts might translate to 1.5 fold change in concentration. If the authors wish to make claims about actual concentrations, they should express their data as such. See technical issues. The reviewers believe these results are probably correct, however, without independent experiments and proper quantification, there is no way to be sure. The authors could in fact be wrong.

Page 14. "These findings reveal a potentially detrimental accumulation of lysine...". This is incorrect and not backed up by the data. The authors can successfully rescue the mutants with homoserine

and threonine, and therefore the toxicity of lysine accumulation does not exist. And that lack of toxicity is partially caused by the lysE export. If there was toxicity due to accumulation of lysine, homoserine and threonine would not be able to rescue the mutants.

Page 15. "Threonine is the main feedback regulator of the aspartate pathway in...". These results are largely confirming work that has been done with purified enzymes (Stadtman, E. R. 1963; Patte, J. C., 1967; Yang, Q. 2011), pointing to the exact same result and with other bacteria.

Page 16. "A lysine relief valve mechanism ensures maintenance". This is only toxic in the presence of dipeptide uptake (Erdmann, A., 1993; Bellmann, A. 2001), which is clearly not the case here. Again, we think the authors are making a huge leap, interpreting their lysine accumulation, which might not be correct, pending conversion of their data to molar scale, as toxic. There is no evidence for that until now from their experiments, or in the TB literature, contrary to the work cited in *C. glutamicum*.

Page 17. All the work with lysine and arginine analogues is weak, as proof of lysine toxicity. To us, all the results obtained indicate that as expected, the two toxic analogues are toxic, because of their incorporation into proteins previously demonstrated in other species, and that they are transported out by LysE. This toxicity is mechanistically unrelated to lysine accumulation, which again has not been formally proved by this work.

Page 17. "Taken together, these data show that *Mtb* utilizes lysine export and degradation pathways as a metabolic relief valve to prevent disruption of aspartate pathway homeostasis and maintain essential precursor production." This statement is not supported by the data in the manuscript. What the authors are probably seeing is carbon re-routing, without a detrimental effect. No lysine toxicity results is presented in this manuscript, yet that is the main conclusion.

Page 18. Results obtained during mouse infection need independent verification. But overall they are not surprising. As now we have seen with several other amino acid auxotrophs in *Mtb*, these lead to potent sterilization in mice. The authors have already demonstrated that for methionine, leucine, and lysine auxotrophs before.

Reviewer #2 (Remarks to the Author):

In this submission, Michael Berney and colleagues investigate the consequences of disrupted aspartate metabolism for mycobacterial growth and pathogenesis. Applying a panel of targeted gene-deletion and regulable gene-depletion mutants in phenotypic, metabolomic, and transcriptomic assays, the authors present evidence supporting the importance of this pathway in vitro and in a mouse model, as well as revealing a central role for lysine accumulation in the rapid death of threonine and homoserine auxotrophs.

Major comments:

1. The text refers throughout to “persisters” in the context of antibiotic tolerance and, equally, organisms which are able to sustain long-term host infection; similarly, “persistence” is used in both contexts. While linguistically correct, this terminology is very confusing scientifically – through no fault of the authors – and represents a confusion that has been propagated owing to the use of these terms in both drug susceptibility and mycobacterial pathogenesis studies. My preference would be for the authors to define up front what they mean by these terms (perhaps substituting “persistent” with “tolerant” for drug susceptibility phenotypes), and to use them consistently thereafter throughout the manuscript.
2. The authors make very confident statements on a handful of occasions; for example, that “metabolic flux through the aspartate pathway needs to be tightly regulated” (L78). Where is the evidence for this? Does this regulation entail transcriptional, translational, and/or metabolite level control, or perhaps any of these? In the absence of compelling evidence (e.g., examples of cellular collapse owing to dysregulation) so definite a statement should be avoided.
3. Similarly, there are several phrases which are hyperbolic and appear better suited to grant applications. For example:
 - a. “advanced mycobacterial genetics” (L85); does this refer to the use of the ATc-dependent expression system which is now in standard use throughout the field?
 - b. “unique combination of feedback control, overflow metabolism, and catabolic action” (L449)
4. The authors interpret the data in Figure 1 as indicating that the thrB mutant was “killed rapidly” (L275) and that the thrA deletion was “bactericidal” (L278). Are these inferences based solely on the OD and CFU data? If so, did the authors confirm cidalty (e.g., via microscopy and/or flow cytometry) or is this conclusion based only on the inability to recover CFU? See also L394.
5. Why were transcriptomic analyses conducted at different time-points for the thrB (0, 1, 2 days) versus thrA (0, 2, 4, 7 days) mutant? Was this based on a scientific rationale, or did it simply take advantage of available datasets?
6. In Figure 3, the authors present data in support of extracellular lysine accumulation in the thrA mutant; where are the comparable data for the thrB mutant? This is important since the ensuing discussion implies equivalent phenotypes for these two auxotrophs (see L336-7, for example).

7. Continuing from the above: to strengthen the inference that aspartate kinase is regulated by threonine (and not lysine), the authors supplemented the thrA (Fig. 3) and thrB (Fig. 4) mutants with threonine, reducing lysine accumulation relative to the unsupplemented strains. This is a tricky experiment: (i) how can the authors separate this observation from their own claim that threonine fully complements growth of the thrB mutant (Fig. 1B, C) and retards death in the thrA mutant (Fig. 1D, E); and (ii) how do they reconcile reduced lysine accumulation in the thrA mutant with the observation that threonine supplementation is not able to alleviate cidality in that mutant?
8. The authors utilize three toxic amino acid analogs which are claimed to disrupt metabolism of the corresponding amino acids (histidine, lysine, arginine). Given the centrality of the resulting observations (Fig. 5) in the argument for lysine and arginine export as a detoxification mechanism, it is essential that MIC data are presented to illustrate that supplementation of growth media with any of the canonical amino acids alleviates the potency of the corresponding toxic analog. In the absence of these data, it is very difficult to make any assumptions about specificity.
9. Continuing from the above: have the authors confirmed in their metabolomic analyses that export of lysine and/or arginine is reduced in a lysG and/or lysE mutant? This should be readily apparent in endo- and exo-metabolomes, and is critical to support the claim made in L485-6.
10. Panel E in Fig.5 appears to be “cropped”; that is, more concentrations need to be added to extend the dataset in the early (100% growth) phase of the Hill curve.
11. The thrA/lysE double mutant would strengthen the authors’ conclusions significantly but appears to be lethal; have the authors attempted to construct a lysE hypomorph in the thrA background?
12. Continuing from the above, it seems strange that the authors utilized the Asd-T38 mutant in mouse experiments and not a lysE mutant. The rationale for including Asd is not at all clear – especially given that this mutant is likely to phenocopy the thrA strain given their locations at subsequent steps in the pathway (Fig. 1) and does not appear at all in the manuscript up to this point.
13. Continuing from the above: the decision not to include a thrB mutant also seems especially strange given the argument in Figs. 1-4 that differentiating the thrA from thrB phenotypes is valuable owing to their different capacities for chemical complementation with threonine and methionine.
14. Comparing panels D, E and F of Fig. 6, there is a large discrepancy in the CFU/lung values achieved by the asd-T38 mutant at week 3 ($\sim 5 \times 10^3$) versus the thrA-DUC ($\sim 1 \times 10^5$) and metX-DUC ($\sim 5 \times 10^4$) mutants at the same 21-day time-point. Is the asd-T38 mutant attenuated even in the absence of ATc (doxycycline) induction?
15. Given the impact of aberrant aspartate metabolism on the cell wall (L457) and the proposal that lysine serves as key alarmone (L475), have the authors investigated the sensitivity of the thrA and/or thrB mutant to known anti-TB agents? Similarly, what is the impact of lysine or threonine supplementation on the susceptibility of wild-type bacilli to known drugs?

16. Although it is only a cartoon, the depiction of LysE as spanning a membrane which separates intracellular from extracellular compartments is too reductive in the context of an organism with as complex a cell wall structure as Mtb.

Minor comments:

1. L41 “led” or “leads” would be preferable
2. L70 The phrase “disruption of methionine biosynthesis in Mtb is required” does not make sense
3. L108 “lysE” [italics]

Reviewer #3 (Remarks to the Author):

This is an elegant study from the Berney lab, which combines genetics, metabolomics and transcriptomics to study the consequences of genetically interrupting the aspartate pathway in *M. tuberculosis*.

The authors make a good case that enzymes involved in this pathway should be considered as potential targets for the development of novel drugs to treat tuberculosis. Druggability of the respective enzymes remains to be demonstrated, but work addressing this question is beyond the scope of the work reported in this manuscript.

The metabolite analyses identified a lysine relief valve as an interesting mechanism to maintain lysine homeostasis. The underlying principle of metabolite expulsion is likely used more generally by *M. tuberculosis* and will likely stimulate further studies.

The manuscript is well written, the experiments were well executed and, for the most part, well controlled.

However, Figure 6 needs to be accompanied by data on histopathology to assure that a reduction in CFUs is accompanied by a reduction of lung pathology.

Furthermore, I am struck by the data for delta-*asd*-T38 shown in Fig 6D/G. This mutant seems to be eliminated much more quickly than the other two mutants. In the most extreme case, titers were apparently reduced by ~5 orders of magnitude in a relatively brief period of time (Fig 6D, doxy week 6). Can one rule out that carryover of doxycycline from the infected organ to the outgrowth plates contributed to this effect?

Dr. Michael Berney
Albert Einstein College of Medicine
Department of Microbiology and Immunology
Price Center for Genetic and Translational Medicine
Bronx, NY 10461-2303
Telephone: (718) 678-1030

April 13th, 2019

Dear Referees,

We would like to thank you for your time and very helpful comments that enabled us to improve our manuscript, strengthen the story and make it more concise.

We have conducted a substantial amount of additional experiments (we added 6 new figures to the supplement including new animal data) that address the reviewers concerns and have rewritten major parts of the manuscript.

Please find below our point-by-point responses (in blue).

Sincerely,

Assistant Professor
Microbiology and Immunology

Reviewers' comments:

Reviewer #1 (Remarks to the Author):

This manuscript describes work towards the characterization of threonine/isoleucine/methionine auxotrophs in *Mycobacterium tuberculosis*. Unfortunately, by technical and scientific reasons explained below the reviewers feel that this manuscript needs significant work to pass peer-review in Nature Communications or other high profile journals. Mainly, there is very little novelty in the results presented.

We strongly disagree with this notion of “very little novelty”.

A major constraint for developing new TB drugs is the limited number of validated targets that allow eradication of persistent infections. Amino acid biosynthesis in *M. tuberculosis* is gaining increasing interest as a drug target space (Zhang *et al.* 2013 *Cell*, Wellington *et al.* 2017 *Nat. Chem. Biol.*, Berney *et al.* 2015 *PNAS*, Tiwari *et al.* 2018 *PNAS*). However, considerable skepticism still remains in the field whether amino acid biosynthesis is indeed a good target space mainly due to a lack of information about its vulnerability during persistent infection. Most other

bacterial pathogens are able to scavenge amino acids from the host, yet *Mtb* seems to represent an exception in this regard. Our study is the first to evaluate the vulnerability of amino acid biosynthesis during different stages of infection. We discovered that *Mtb* is unable to scavenge sufficient amino acids from the host throughout the course of an infection. In all cases we observed clearance of infection even after full bacterial burden was established, pointing to promising candidate enzymes for inhibitor discovery. By studying *Mtb*'s adaptive response to pathway blockage we uncovered a unique metabolic control mechanism that involves an amino acid relief valve (efflux mechanism) and catabolism to balance pathway homeostasis and maintain essential precursor production. *Mtb* is an exception in the prokaryotic world, in that it does not allosterically control aspartate kinase by lysine. Instead, we discovered an arginine/lysine exporter that acts as a relief valve to counter flux imbalance and a degradation pathway that is unique to mycobacteria. The exporter could play an important immuno-metabolic role in shaping *Mtb*'s niche during infection as it is a major epitope in latent TB infection in humans and a dominant target of IL-2 secreting memory T cells (Gideon *et al.* 2010 *PLoS Pathog*).

Because our findings (1) fundamentally answer questions about mycobacterial persistence metabolism in the host, (2) highlight the vulnerability of amino acid biosynthesis during chronic infection, and (3) elucidate the metabolic control mechanism in an essential anabolic pathway, we believe that our manuscript is novel and has broad appeal.

It appears that the referees concern about novelty stems from the fact that some experimental outcomes were "expected", e.g. Tn screens predict the genes studied here to be essential. However, the referee seems to ignore the fact that Tn screens are unreliable predictors of *in vivo* essentiality and cannot differentiate between lethality, bacteriostasis, slow growth phenotypes or conditional essentiality, yet this is crucial information for assessing druggability. Also, due to technical limitations (not enough amino acid in the growth medium), Tn screens predicted MetX, ThrA and ThrB to be essential, although these enzymes are only conditionally essential in the absence of their pathway end products. Even more importantly, Tn screens to date have never addressed the question of whether threonine, methionine, lysine or any other amino acid can be scavenged from the host during chronic infections. Since *M. tuberculosis* is an intracellular pathogen, essentiality and druggability of targets have to be tested during chronic infection in order to prevent false targets being taken forward for drug development.

Technical issues-

Q1. All mass spec data is reported as: abundance, normalized abundance and fold-change. While this style of presenting data is very popular with microarrays, it has no direct use in metabolism. The authors must be aware that different compounds ionize differently and that needs to be determined with standard curves. Through the manuscript the authors refer to abundance, relative abundance and so on, without really communicating with clarity what are they referring to.

R1. Thank you for pointing this out. With abundances we refer to the raw intensity values for the area under the curve for each metabolite peak. With fold-changes we refer to normalization to t_0 . We made this clearer in the manuscript (Line 166-170).

We routinely run standards of the targeted metabolites at different concentrations to determine the linear range. As long as the levels measured are in the linear range for the specific metabolite detection, comparison of raw counts or fold changes, are acceptable representations (for example, see Science DOI: 10.1126/science.aau8959, Figure 2b, or Nature Microbiology

DOI:10.1038/nmicrobiol.2017.84, Figure 3, Nature Methods doi:10.1038/nmeth.3584, Figure 1). Our data presentation is therefore consistent with other high impact studies.

Below you will find a direct comparison of different representations of the four main compounds that are important to our manuscript (lysine, aminoadipate, diaminopimelate and acetyl-lysine).

Line one: raw counts,

Line two: fold-changes (normalized to t0) based on raw counts,

Line 3: conc. in μM based on standard curves,

Line 4: Fold-changes calculated based on μM conc.

As is very evident from this data, the following statement from the manuscript is strongly justified: “In accordance with the upregulation of genes for lysine disposal, the $\Delta thrA$ mutant in media lacking homoserine displayed a significant increase (up to 100-fold) of all identifiable metabolites in the lysine biosynthesis pathway (Figure 3)”.

Q2: Another major issue that precludes acceptance of this study and the claims made by the authors is the lack of independent experiments. All data presented appears to have been generated once, with biological replicates. The reviewers have no trust that the results are in fact reproducible. The authors must show that their data are reproducible by carrying truly independent experiments (different day, different cells, different media, etc). This is particularly important for metabolic experiments as well as microbiological experiments.

R2: We are confused by the comments of the referee and we have clarified what we mean by biological replicates. We always measure biological replicates (i.e. > 1) and repeat experiments

on different days. Please find below an independent repeat experiment of the metabolomic data (again showing the four main metabolites), error bars represent standard deviations from 3 biological replicates. We have now added the following sentence to the figure legends: "All values are the average of three biological replicates ($n = 3$) \pm s.d. and are representative of a minimum of two independent experiments. ** p-value <0.01 in student t-test."

Regarding the mouse data of the conditional knockdowns, we performed additional experiments to test if KD's worked in mice (see below) before we did the full experiment that ran for 300 days involving hundreds of mice. Repetition of full KD experiments with such clear phenotypes for multiple enzymes of the same pathway are not customary in the field (please see published work in *Nature Com* or *PNAS*: DOI: 10.1038/ncomms1473, 10.1073/pnas.1315860110). Animal welfare, time and financial constraints argue against unnecessary animal experiments. In addition to the KD study, we show data for deletion mutants in BL6 and SCID mice (see supplementary Figures S9 and S10). Taken together, we argue that our data strongly supports our conclusions and is again consistent with other high impact studies using animal models.

Other Major Issues:

Q3.1:Page 12/13. The authors have confirmed that threonine and homoserine auxotrophy is lethal to *Mtb*. Although, these exact experiments have not been carried out before, *thrA*, *thrB* are essential in *Mtb* (Sasseti CM et al. 2003; Griffin JE et al. 2011; DeJesus MA et al. 2017). These results are expected based on the vast data on transposon mutants and the current understanding of amino acid biosynthesis in *Mtb*.

R3.1. We would strongly disagree with the referee about interpretations from transposon mutant screens. Such screens have been valuable indicators of potential *in vitro* gene essentiality but they have serious limitations. They are unreliable predictors of *in vivo* essentiality and cannot

differentiate between lethality, bacteriostasis, slow growth phenotypes or conditional essentiality, yet this is crucial information for assessing druggability (please see: doi.org/10.1073/pnas.1315860110 or doi.org/10.1073/pnas.1018301108). Even more importantly, Tn screens to date have never addressed the question if threonine, methionine, lysine or any other amino acid can be scavenged from the host during chronic infections. This is in part due to the fact that media to isolate Tn mutants are not supplemented with high enough amino acid concentrations to select for amino acid auxotrophs, hence the starting libraries that go into animals are already devoid of such mutants. Tuberculosis is a deadly disease and finding drug targets and corresponding drugs is of major importance, so the science should not stop at Tn screens.

Q3.2: In addition, the authors have confirmed that Ile does not rescue Thr auxotrophy. This is also expected based on the number of irreversible steps after Thr.

R3.2. We agree that this might be expected, yet this was not the major focus of the experiment. We needed to investigate if the absence of isoleucine rather than threonine is responsible for the cell killing phenotype seen in the threonine auxotroph, because isoleucine biosynthesis is dependent on threonine. We have revised this sentence and removed the part on irreversibility.

Q4: Page 13/14. As the authors did not express the metabolomics data as concentration (molar or g/mL) the authors cannot really conclude anything with regards to accumulation. 10-fold change in lysine ion counts might translate to 1.5 fold change in concentration. If the authors wish to make claims about actual concentrations, they should express their data as such. See technical issues. The reviewers believe these results are probably correct, however, without independent experiments and proper quantification, there is no way to be sure. The authors could in fact be wrong.

R4. This comment has largely been addressed in points R1 and R2. We have converted the raw values to concentrations and the message remains the same.

Furthermore, we have added dynamic data of extracellular lysine accumulation (Figure S2) showing up to 160 μM of lysine in the cell-free supernatant (Line 316-319). We have also calculated the intracellular lysine concentration to be around 10 mM in the *thrA* mutant (using an average cell volume of 8.4 μm^3), which is in the same range as the K_m determined for LysE in *C. glutamicum* (20 mM). This information, however, will not go into the manuscript, as calculations of intracellular concentration hinge on too many unreliable factors (i.e. cell volume variation, cell number estimation, metabolite extraction efficiency).

Q5.1: Page 14. “These findings reveal a potentially detrimental accumulation of lysine...”. This is incorrect and not backed up by the data. The authors can successfully rescue the mutants with homoserine and threonine, and therefore the toxicity of lysine accumulation does not exist. And that lack of toxicity is partially caused by the lysE export. If there was toxicity due to accumulation of lysine, homoserine and threonine would not be able to rescue the mutants.

R5.1: We feel that this comment indicates a misunderstanding of our proposed model. Our results unequivocally demonstrate that the $\Delta thrA$ strain accumulates lysine pathway intermediates and that replacement of threonine, while unable to rescue growth, decreases the intracellular

abundance of lysine pathway intermediates to WT levels and delays killing. Together with the fact that threonine is a feedback inhibitor of *Mtb* aspartate kinase *in vitro* (Yang, Q. 2011), we conclude that accumulation of lysine, or its catabolites, and any potential toxicity would be reversed by addition of threonine likely through feedback inhibition of Ask.

Q5.2: Page 15. “Threonine is the main feedback regulator of the aspartate pathway in...”. These results are largely confirming work that has been done with purified enzymes (Stadtman, E. R. 1963; Patte, J. C., 1967; Yang, Q. 2011), pointing to the exact same result and with other bacteria.

R5.2: We discuss our findings in the context of exactly those publications (see line 328) in the original manuscript (now L341). The key point of this paragraph is that *M. tuberculosis* controls aspartate kinase differently than the majority of other prokaryotes i.e. there is no allosteric feedback of lysine or methionine, but only by threonine. Our results clearly show that threonine is the main regulator of aspartate kinase in whole cells, which is an important finding for the overall conclusions of our paper. It would be wrong to just assume this to be the case based on enzyme assays on the purified protein as such assays do not always accurately predict *in vivo* specificity and impact on activity. Importantly, the referenced study by Yang et al. did not assess if any other amino acids (e.g. methionine) controlled activity of AK nor was it investigated if these results translate to whole cell physiology. We have added this to the manuscript (L345).

Q6: Page 16. “A lysine relief valve mechanism ensures maintenance”. This is only toxic in the presence of dipeptide uptake (Erdmann, A., 1993; Bellmann, A. 2001), which is clearly not the case here. Again, we think the authors are making a huge leap, interpreting their lysine accumulation, which might not be correct, pending conversion of their data to molar scale, as toxic. There is no evidence for that until now from their experiments, or in the TB literature, contrary to the work cited in *C. glutamicum*.

R6: The title reads: “*A lysine relief valve mechanism ensures maintenance of lysine homeostasis*”. There is no mention of toxicity. We list this as a possible reason for why lysine is exported, but balanced growth and maintenance of correct precursor allocation might be other more plausible explanations. The main message is that we characterized a bacterium that has no lysine-AK feedback loop, but instead uses a mycobacterium-specific degradation pathway and lysine export to balance metabolite levels in a biosynthetic pathway. We have now toned down the language throughout the manuscript, to make it clear that toxicity from metabolite accumulation in the lysine pathway is a possibility but not a certainty.

It is also important to mention here, that a natural, physiological role of LysE in WT *C. glutamicum* or in any other WT bacterium has not been determined, probably due to the focus on lysine production for industrial applications. Our work gives a plausible explanation why an intracellular pathogen like *Mtb* has maintained a functional copy of this exporter as well as a rare lysine degradation pathway. To our knowledge, our work is the first to show that *M. tuberculosis* has the capacity to export an amino acid. This was also recognized and supported by Reviewer 3.

Q7:Page 17. All the work with lysine and arginine analogues is weak, as proof of lysine toxicity. To us, all the results obtained indicate that as expected, the two toxic analogues are toxic, because of their incorporation into proteins previously demonstrated in other species, and that

they are transported out by LysE. This toxicity is mechanistically unrelated to lysine accumulation, which again has not been formally proved by this work.

R7: These experiments were not designed to prove lysine toxicity. These experiments were performed to test if LysE is indeed a lysine/arginine exporter in *M. tuberculosis*, to substantiate our claim that in cases of increased lysine abundance in the cell, this permease can act as a relief/overflow valve to export excess lysine. Please also see R8 to reviewer 2.

Page 17. “Taken together, these data show that Mtb utilizes lysine export and degradation pathways as a metabolic relief valve to prevent disruption of aspartate pathway homeostasis and maintain essential precursor production.” This statement is not supported by the data in the manuscript. What the authors are probably seeing is carbon re-routing, without a detrimental effect. No lysine toxicity results is presented in this manuscript, yet that is the main conclusion.

R8: This sentence discusses our observation that when lysine accumulates in the *thrA* mutant, it is exported by LysE and degraded by the amino adipate pathway. This happens in the absence of lysine feedback-control on aspartate kinase that is found in most other prokaryotes. The main finding of this part of the study is an alternative mechanism of dealing with fluctuations in the lysine biosynthesis branch of the aspartate pathway. Toxicity is not the main conclusion but a possibility worthy of discussion. We have now toned down the language throughout the manuscript, to make it clear that toxicity from metabolite accumulation in the lysine pathway is just a possibility.

Page 18. Results obtained during mouse infection need independent verification. But overall they are not surprising. As now we have seen with several other amino acid auxotrophs in Mtb, these lead to potent sterilization in mice. The authors have already demonstrated that for methionine, leucine, and lysine auxotrophs before.

R9: To our knowledge, there is not a single published study that shows that amino acid auxotrophy is lethal during chronic infection of Mtb. We state this in the introduction as follows: Line 65-68: “However, none of the amino acid biosynthetic pathways of Mtb have yet been shown to be essential during chronic infection and it is largely unknown if such building blocks can be scavenged from the host during persistence.”

The studies mentioned by the reviewer do not address the specific question if bacteria that have established an infection, are susceptible to target inhibition. Lethality during chronic infection is a major outcome of our study and extremely important not only for drug discovery, but also for understanding the role of nutritional immunity and the metabolic environment of *Mtb* during an established infection. It strongly indicates that these enzymes, when targeted by drugs, are vulnerable during chronic infection, potentially leading to sterilization and at the same time show for the first time that these amino acids cannot be scavenged during an established infection. Since *M. tuberculosis* is an intracellular pathogen, essentiality and druggability of targets have to be tested during chronic infection in order to prevent false targets being taken forward for drug development. Inability to grow on a plate (e.g. in transposon screens) or to mount an initial infection are not good proxies for essentiality during chronic *Mtb* infections. Please see R2 regarding independent verification.

Reviewer #2 (Remarks to the Author):

In this submission, Michael Berney and colleagues investigate the consequences of disrupted aspartate metabolism for mycobacterial growth and pathogenesis. Applying a panel of targeted gene-deletion and regulable gene-depletion mutants in phenotypic, metabolomic, and transcriptomic assays, the authors present evidence supporting the importance of this pathway in vitro and in a mouse model, as well as revealing a central role for lysine accumulation in the rapid death of threonine and homoserine auxotrophs.

Major comments:

1. The text refers throughout to “persisters” in the context of antibiotic tolerance and, equally, organisms which are able to sustain long-term host infection; similarly, “persistence” is used in both contexts. While linguistically correct, this terminology is very confusing scientifically – through no fault of the authors – and represents a confusion that has been propagated owing to the use of these terms in both drug susceptibility and mycobacterial pathogenesis studies. My preference would be for the authors to define up front what they mean by these terms (perhaps substituting “persistent” with “tolerant” for drug susceptibility phenotypes), and to use them consistently thereafter throughout the manuscript.

R1: We thank the reviewer for this suggestion. We have changed the introduction entirely so that no confusion about these terms should arise. See L49-57.

2. The authors make very confident statements on a handful of occasions; for example, that “metabolic flux through the aspartate pathway needs to be tightly regulated” (L78). Where is the evidence for this? Does this regulation entail transcriptional, translational, and/or metabolite level control, or perhaps any of these? In the absence of compelling evidence (e.g., examples of cellular collapse owing to dysregulation) so definite a statement should be avoided.

Direct evidence for Similarly, there are several phrases which are hyperbolic and appear better suited to grant applications. For example:

- a. “advanced mycobacterial genetics” (L85); does this refer to the use of the ATc-dependent expression system which is now in standard use throughout the field?
- b. “unique combination of feedback control, overflow metabolism, and catabolic action” (L449)

R2: We have replaced these passages with more pragmatic wording. See L71-74, L80, L462, and have removed the hyperbole in other places.

4. The authors interpret the data in Figure 1 as indicating that the thrB mutant was “killed rapidly” (L275) and that the thrA deletion was “bactericidal” (L278). Are these inferences based solely on the OD and CFU data? If so, did the authors confirm cidality (e.g., via microscopy and/or flow cytometry) or is this conclusion based only on the inability to recover CFU? See also L394.

R4: Yes, these results are based on CFU data as this is still the gold standard for viability testing in microbiology and drug discovery. However, we do acknowledge that there are certain physiological conditions where bacterial strains might survive without showing growth on an agar plate. Cultivation-independent viability methods (mentioned by the reviewer) are also not definitive (e.g. doi:10.1016/j.mimet.2017.09.011, doi: 10.1128/AEM.02750-06). In addition, cultivation-

independent viability methods for *Mtb* are still poorly developed and not quantitative because of the problem with cell clumping.

5. Why were transcriptomic analyses conducted at different time-points for the *thrB* (0, 1, 2 days) versus *thrA* (0, 2, 4, 7 days) mutant? Was this based on a scientific rationale, or did it simply take advantage of available datasets?

R5: This was based on scientific rationale. Chronologically we studied the *thrA* strain first and noticed that the main transcriptomic and metabolic changes are happening in the first 2 days. Hence we focused on this timeframe for the rest of the study. We have added this information to the methods section (L99-101)

6. In Figure 3, the authors present data in support of extracellular lysine accumulation in the *thrA* mutant; where are the comparable data for the *thrB* mutant? This is important since the ensuing discussion implies equivalent phenotypes for these two auxotrophs (see L336-7, for example).

R6. We have also analyzed the $\Delta thrB$ exometabolome and found increased lysine concentrations albeit to a lower extent than in $\Delta thrA$ (see figure S4), which is consistent with the fact that in $\Delta thrB$ part of the increased flux through the pathway is diverted to homoserine (see R7). We have added this data (Figure S4) to the manuscript and discussed it in L325-327.

7. Continuing from the above: to strengthen the inference that aspartate kinase is regulated by threonine (and not lysine), the authors supplemented the *thrA* (Fig. 3) and *thrB* (Fig. 4) mutants with threonine, reducing lysine accumulation relative to the unsupplemented strains. This is a tricky experiment: (i) how can the authors separate this observation from their own claim that threonine fully complements growth of the *thrB* mutant (Fig. 1B, C) and retards death in the *thrA* mutant (Fig. 1D, E); and (ii) how do they reconcile reduced lysine accumulation in the *thrA* mutant with the observation that threonine supplementation is not able to alleviate cidality in that mutant?

R7. We agree with the reviewer that the situation is very complex, as we are possibly dealing with multiple factors that contribute to cell death. Starvation for an amino acid, cofactor of nutrient in many cases is bacteriostatic or only slowly bactericidal, hence rapid cell death, like seen in $\Delta thrA$ and $\Delta thrB$, is often an indicator of accumulation of toxic intermediates that accelerate killing.

(i) Our data strongly suggests that killing is multifactorial in the case of both $\Delta thrA$ and $\Delta thrB$. In $\Delta thrA$, death likely occurs through a combination of (1) loss of proteinogenic methionine, (2) loss of proteinogenic threonine, and (3) accumulation of lysine pathway intermediates. We show that in methionine supplemented medium the rate of killing is retarded, indicating that there is a difference between the loss of (1) and the loss of (2) and (3).

In $\Delta thrB$, death likely occurs through a combination of (1) homoserine accumulation, (2) loss of proteinogenic threonine, and (3) lysine pathway intermediate accumulation. In our analysis of the $\Delta thrB$ mutant data we found that $\Delta thrB$ but not $\Delta thrA$ or $\Delta metX$ mutants accumulates homoserine (see new figure S3). Homoserine had been proposed to be a toxic amino acid for *Mtb* (O'Barr 1971) and we have confirmed this in the *Mtb* complex strain *M. bovis* BCG (see new figure S5). Originally, we had not included this data, because differentiation of homoserine and threonine by

mass spectrometry is not conclusive due to exact same m/z value, almost identical retention time (0.2 min difference) and almost identical fragmentation pattern. However, due to the importance of these results, we have now worked it into the manuscript (see Line 319-333) and we give a detailed description of how we identify homoserine in the methods section (Line 171-179). The two figures were added to the supplement as new Figures S3 and S5.

The main message is that ThrA and ThrB represent promising drug targets and the absence of threonine revealed a rare disposal mechanism (export and degradation) for lysine accumulation in *Mycobacterium tuberculosis* that is of interest to the field.

ii) Addition of threonine to the $\Delta thrA$ mutant slows down cell killing by replacing proteinogenic threonine (2) and/or lysine intermediate accumulation (3), the rest of the cidality stems from the loss of proteinogenic methionine and S-adenosylmethionine (1) (e.g. *metX* mutation is cidal).

8. The authors utilize three toxic amino acid analogs which are claimed to disrupt metabolism of the corresponding amino acids (histidine, lysine, arginine). Given the centrality of the resulting observations (Fig. 5) in the argument for lysine and arginine export as a detoxification mechanism, it is essential that MIC data are presented to illustrate that supplementation of growth media with any of the canonical amino acids alleviates the potency of the corresponding toxic analog. In the absence of these data, it is very difficult to make any assumptions about specificity.

R8: This is a great suggestion – thank you. We have now performed this experiment with canavanine and arginine. Figure S6 shows that 50 $\mu\text{g/ml}$ arginine alleviates canavanine toxicity in the $\Delta lysE$ and $\Delta lysG$ mutant and brings their sensitivity back to WT levels. The results with canavanine and arginine nicely address the reviewers concern and add great value to the paper. We discuss these findings in Line 380-382 and added the new Figure S6 to the supplementary material. (Methods: line 257-258)

9. Continuing from the above: have the authors confirmed in their metabolomic analyses that export of lysine and/or arginine is reduced in a *lysG* and/or *lysE* mutant? This should be readily apparent in endo- and exo-metabolomes, and is critical to support the claim made in L485-6.

R9. This experiment could only be done in a $\Delta thrA \Delta lysE$ double mutant, because the *LysE* mutant itself does not accumulate lysine to high enough concentrations under the conditions tested so far. This was the justification for performing the toxic analog experiments and is now supported further by Figure S6.

10. Panel E in Fig.5 appears to be “cropped”; that is, more concentrations need to be added to extend the dataset in the early (100% growth) phase of the Hill curve.

R10. We have repeated the experiment and show more data points (see revised Figure 5E.)

11. The *thrA/lysE* double mutant would strengthen the authors’ conclusions significantly but appears to be lethal; have the authors attempted to construct a *lysE* hypomorph in the *thrA* background?

R11. We have not attempted to make the *lysE* hypomorph in the $\Delta thrA$ background, but this remains an experiment for future work in this area.

12. Continuing from the above, it seems strange that the authors utilized the Asd-T38 mutant in mouse experiments and not a *lysE* mutant. The rationale for including Asd is not at all clear – especially given that this mutant is likely to phenocopy the *thrA* strain given their locations at subsequent steps in the pathway (Fig. 1) and does not appear at all in the manuscript up to this point.

R12. The reason for including the Asd data was chronological as we decided to make a knockdown of Asd before the rest of this manuscript developed. We do agree with the reviewer and have removed the Asd-T38 mutant data from the paper. Instead, we have now performed a further mouse infection experiment with the $\Delta thrB$ mutant, which fits better into the narrative of this manuscript. As can be seen in new Figure S9, the $\Delta thrB$ mutant is avirulent in BL6 mice.

13. Continuing from the above: the decision not to include a *thrB* mutant also seems especially strange given the argument in Figs. 1-4 that differentiating the *thrA* from *thrB* phenotypes is valuable owing to their different capacities for chemical complementation with threonine and methionine.

R13. We have removed the Asd-Kd strain from this study and instead have now performed a mouse infection experiment with the $\Delta thrB$ mutant (see comment R12). As can be seen in figure S10, the $\Delta thrB$ mutant is avirulent in BL6 mice.

14. Comparing panels D, E and F of Fig. 6, there is a large discrepancy in the CFU/lung values achieved by the *asd*-T38 mutant at week 3 ($\sim 5 \times 10^3$) versus the *thrA*-DUC ($\sim 1 \times 10^5$) and *metX*-DUC ($\sim 5 \times 10^4$) mutants at the same 21-day time-point. Is the *asd*-T38 mutant attenuated even in the absence of ATc (doxycycline) induction?

R14. Our *in vitro* experiment (previously Figure 6A) indicates that the *asd*-T38 strain is not attenuated in the absence of ATc. However, this could be different under *in vivo* conditions. As stated above, we have removed the *asd*-T38 data from the manuscript as it is tangential to the story.

15. Given the impact of aberrant aspartate metabolism on the cell wall (L457) and the proposal that lysine serves as key alarmone (L475), have the authors investigated the sensitivity of the *thrA* and/or *thrB* mutant to known anti-TB agents? Similarly, what is the impact of lysine or threonine supplementation on the susceptibility of wild-type bacilli to known drugs?

R15. This is an interesting question and should be addressed in future experiments to determine potential synergisms or antagonism of *thrA/thrB/metA* inhibition with current anti-tuberculosis drugs.

16. Although it is only a cartoon, the depiction of LysE as spanning a membrane which separates intracellular from extracellular compartments is too reductive in the context of an organism with as complex a cell wall structure as *Mtb*.

R16. We agree that the mycobacterial cell wall is more complex than depicted in this cartoon, but we argue that abstraction is useful in a cartoon to make the point that lysine and arginine are exported. In order to not confuse the reader, we have replaced the lipid bilayer, by a plain bar that represents the entire *Mtb* plasma membrane and cell wall (see revised Figure 2).

Minor comments:

1. L41 “led” or “leads” would be preferable. Corrected
2. L70 The phrase “disruption of methionine biosynthesis in *Mtb* is required” does not make sense Corrected
3. L108 “*lysE*” [italics] Corrected

Reviewer #3 (Remarks to the Author):

This is an elegant study from the Berney lab, which combines genetics, metabolomics and transcriptomics to study the consequences of genetically interrupting the aspartate pathway in *M. tuberculosis*.

The authors make a good case that enzymes involved in this pathway should be considered as potential targets for the development of novel drugs to treat tuberculosis. Druggability of the respective enzymes remains to be demonstrated, but work addressing this question is beyond the scope of the work reported in this manuscript.

The metabolite analyses identified a lysine relief valve as an interesting mechanism to maintain lysine homeostasis. The underlying principle of metabolite expulsion is likely used more generally by *M. tuberculosis* and will likely stimulate further studies.

The manuscript is well written, the experiments were well executed and, for the most part, well controlled.

Q1. However, Figure 6 needs to be accompanied by data on histopathology to assure that a reduction in CFUs is accompanied by a reduction of lung pathology.

R1. Gross lung pathology and histopathology have now been added to the main text (see revised Figure 6). Methods L246-248, Results: L432-437

Q2. Furthermore, I am struck by the data for delta-*asd*-T38 shown in Fig 6D/G. This mutant seems to be eliminated much more quickly than the other two mutants. In the most extreme case, titers were apparently reduced by ~5 orders of magnitude in a relatively brief period of time (Fig 6D, doxy week 6). Can one rule out that carryover of doxycycline from the infected organ to the outgrowth plates contributed to this effect?

R2. All plates were supplemented with the corresponding amino acids. However, as a consequence of comments from reviewer 2 (see R12-14) to make the paper more concise, we have now removed the *Asd* data from this manuscript. Instead we have performed an additional mouse experiment with a \$\Delta thrB\$ mutant (see new Figure S10).

Albert Einstein College of Medicine

Montefiore

** See Nature Research's author and referees' website at www.nature.com/authors for information about policies, services and author benefits

This email has been sent through the Springer Nature Tracking System NY-610A-NPG&MTS

Confidentiality Statement:

This e-mail is confidential and subject to copyright. Any unauthorised use or disclosure of its contents is prohibited. If you have received this email in error please notify our Manuscript Tracking System Helpdesk team at <http://platformsupport.nature.com> .

Details of the confidentiality and pre-publicity policy may be found here

<http://www.nature.com/authors/policies/confidentiality.html>

Privacy Policy | Update ProfileDISCLAIMER: This e-mail is confidential and should not be used by anyone who is not the original intended recipient. If you have received this e-mail in error please inform the sender and delete it from your mailbox or any other storage mechanism. Springer Nature Limited does not accept liability for any statements made which are clearly the sender's own and not expressly made on behalf of Springer Nature Ltd or one of their agents. Please note that Springer Nature Limited and their agents and affiliates do not accept any responsibility for viruses or malware that may be contained in this e-mail or its attachments and it is your responsibility to scan the e-mail and attachments (if any).

Reviewers' comments:

Reviewer #1 (Remarks to the Author):

We thank the authors for trying to address our concerns. Three main problems still persist, that in our view affect the claimed novelty and impact of this manuscript.

1- Lysine accumulation is expected, given the interconnection between threonine and methionine metabolism, as Asp semialdehyde has to go somewhere. Furthermore, the fact that lysine is "secreted" and the lack of rescue by threonine indicates that lysine accumulation is of no consequence, and not the reason behind the phenotype observed. The authors are trying to over-sell this, while the physiology appears to be trivial. We do not understand why this is such a big deal. It is solid work, but the novelty and impact is minimal.

We are really trying to say that if lysine accumulation has no toxic component, it is irrelevant to the biological phenotype the authors claim they are trying to solve.

2- We are not particularly impressed with transposon screens either, however, if the authors' results, with defined genetic deletion strains, are similar to what has been reported in transposon mutagenesis studies there is no novelty there.

3- Finally, we think the authors are over- and mis-interpreting lysine "secretion". The authors have not really demonstrated secretion by lysE and certainly not demonstrated lack of secretion by the lysE deletion strain. All they showed is that there is more lysine out, which could be derived from dead cells or from another transporter.

- Have the authors analyzed other amino acids, unrelated to aspartate/threonine/lysine/arginine/methionine? Is the overall level of metabolites in spent media higher in these mutants?
- Also, the title of figure 5 indicates and the figure shows "co-transport". Are the authors sure lysine and arginine are being co-transported (at the same time)? If yes, which experiments in the paper prove that?
- Furthermore, the experiments with lysE deletion strain are interesting. However, why not show that deletion of the transporter you are claiming is transporting lysine and arginine out, abrogates transport? Without showing that lysine/arginine "secretion" is significantly diminished in

this deletion strain, the authors are omitting an important result. This manuscript cannot be published claiming that lysE is secreting two amino acids without showing that.

- The results presented with amino acid analogs were interpreted by the authors as supporting their statement, however if their hypothesis was correct, and lysE was secreting lysine and arginine, upon deletion, no secretion should be observed. Consequently, intracellular lysine and arginine would increase, which would be protective against toxic analogues. This is the exact opposite of what the authors have. Hence, without data on the diminished secretion of lysine and arginine by the lysE deletion and the increase in toxicity of these amino acid analogs we can only conclude that lysE is not an exporter of these amino acids, as the authors claimed.

Reviewer #2 (Remarks to the Author):

The revised version of this manuscript is a vastly improved document which has largely addressed the major concerns identified in the first round of reviews; the authors have included a significant amount of additional experimental data which strongly support their central claims. Having considered their thorough rebuttal and accompanying manuscript, only the following substantive concern remains:

Q1. Given the heavy reliance in this study on a set of mutants comprising key gene inactivations and complementations – especially for in vivo experiments – it is critical to know whether the authors checked these strains throughout the experimental work (i.e., before and after inoculation/ infection) for the integrity of phthiocerol dimycocerosate (PDIM) biosynthesis (doi:10.1128/JB.00166-10)? This is not only highly relevant to the virulence/pathology phenotypes reported (doi:10.1099/mic.0.029199-0) but, considering the structure and composition of the mycobacterial cell membrane (doi:10.1146/annurev-micro-091014-104121), is almost certain to impact also the respective capacities of these strains for efficient amino acid uptake/export.

Q2. I have also been asked to provide further feedback regarding the concerns raised by Reviewer #1. This reviewer has identified three major concerns with this work, in particular that the inter-related lysine "secretion"/ lysine-arginine co-transport claims are overstated and not adequately supported by the experimental data presented here.

To me, this is the most compelling of the Reviewer #1's arguments and, therefore, a potentially fatal flaw in this work. Consequently, having considered the manuscript and the reviewer's comment in

this regard, I support the demand for further experimental evidence for this. The reviewer has made some reasonable and achievable suggestions to this end:

a. Have the authors analysed other amino acids, unrelated to aspartate/threonine/lysine/arginine/methionine? Is the overall level of metabolites in spent media higher in these mutants?

This is critical, and easily done.

b. Provide at least one piece of experimental evidence proving the claimed co-transport effect.

This is also critical.

These additional experiments will add value and, having considered the reviewer's comments, I agree are essential for publication

Reviewer #3 (Remarks to the Author):

I am satisfied with the authors response to my critique.

Dr. Michael Berney
Albert Einstein College of Medicine
Department of Microbiology and Immunology
Price Center for Genetic and Translational Medicine
Bronx, NY 10461-2303
Telephone: (718) 678-1030

June 18th, 2019

Dear Referees,

We would like to thank you again for your time and helpful comments that enabled us to improve our manuscript.

We have conducted additional experiments and respond point-by-point to your criticisms (in blue).

Sincerely,

Assistant Professor
Microbiology and Immunology

Reviewer #1 (Remarks to the Author):

Q1- Lysine accumulation is expected, given the interconnection between threonine and methionine metabolism, as Asp semialdehyde has to go somewhere. Furthermore, the fact that lysine is "secreted" and the lack of rescue by threonine indicates that lysine accumulation is of no consequence, and not the reason behind the phenotype observed. The authors are trying to over-sell this, while the physiology appears to be trivial. We do not understand why this is such a big deal. It is solid work, but the novelty and impact is minimal.

We are really trying to say that if lysine accumulation has no toxic component, it is irrelevant to the biological phenotype the authors claim they are trying to solve.

R1: The reviewer might have missed this: in Figure 1E we clearly show that addition of threonine considerably slows down killing of the *thrA* mutant (2 logs difference after 10 days), which is a partial rescue. Of course, the lethal consequences of methionine auxotrophy still remains under these conditions, hence the *thrA* mutant is still decreasing in viability.

Our primary conclusion is that ThrA and ThrB are promising drug targets for the development of new anti-mycobacterials and that inactivation of these genes is bactericidal through a multi-factorial killing mechanism, secondarily our data suggests this may include toxic lysine accumulation given its maintenance of lysine disposal and its significant upregulation under threonine starvation. The value of

this work lies in (1) the identification of new drug targets that are essential in both acute and chronic conditions (an important qualifier), and (2) novel characterization of related lysine degradation and export pathways. This latter contribution is particularly important, in spite of definitive proof of lysine toxicity, as it provides greater understanding of lysine catabolic pathways that are unique to mycobacteria and implicated in persistence, cellular stress responses (osmotic, pH, heat-shock, oxidative stress, and infection), macrophage invasion, and protective immunity. This argument has already been made more clearly and elaborately in the manuscript (455-502).

We have included our previous response R7 (Rev 2) from the last revisions because reviewer 2 has already asked about our conclusions regarding lysine accumulation/potential toxicity and we gave a very detailed explanation and interpretation of these observations. We wrote:

“We agree with the reviewer that the situation is very complex, as we are possibly dealing with multiple factors that contribute to cell death. Starvation for an amino acid, cofactor or nutrient in many cases is bacteriostatic or only slowly bactericidal, hence rapid cell death, like seen in $\Delta thrA$ and $\Delta thrB$, is often an indicator of accumulation of toxic intermediates that accelerate cell killing.

(i) Our data strongly suggests that killing is multifactorial in the case of both $\Delta thrA$ and $\Delta thrB$ mutations. In the $\Delta thrA$ mutant, death likely occurs through a combination of (1) loss of proteinogenic methionine, (2) loss of proteinogenic threonine, and (3) accumulation of lysine pathway intermediates. We show that in methionine or threonine supplemented medium the rate of killing is significantly slowed, indicating that there is a major difference between the loss of (1) and the loss of (2) and (3).

In the $\Delta thrB$ mutant, death likely occurs through a combination of (1) homoserine accumulation, (2) loss of proteinogenic threonine, and (3) lysine pathway intermediate accumulation. In our analysis of the $\Delta thrB$ mutant data, we found that $\Delta thrB$ mutant, but not $\Delta thrA$ or $\Delta metX$ mutants, accumulates homoserine (see new figure S3). Homoserine has long been proposed to be a toxic amino acid for *Mtb* (O’Barr 1971) and we have confirmed this in the *Mtb* complex strain *M. bovis BCG* (see new figure S5). Originally, we had not included this data, because differentiation of homoserine and threonine by mass spectrometry is not conclusive due to exact same m/z value, and almost identical retention time (0.2 min difference) and identical fragmentation pattern. However, due to the importance of these results, we have now included it into the manuscript (see Line 319-333). We give a detailed description of how we identify homoserine in the methods section (Lines 171-179). The two figures were added to the supplementary material as new Figures S3 and S5.

The main message here is that in the absence of threonine biosynthesis we have uncovered a hidden disposal mechanism (export and degradation) for lysine accumulation in *Mycobacterium tuberculosis*. Not only will these new observations be of genuine interest to the field, the ThrA and ThrB proteins may hold particular promise as new drug targets for combatting tuberculosis disease.

ii) Addition of threonine to the $\Delta thrA$ mutant slows down cell killing by replacing proteinogenic threonine (2) and/or lysine intermediate accumulation (3), the rest of the toxicity stems from the loss of proteinogenic methionine and S-adenosylmethionine (1) (e.g. *metX* mutation is lethal).”

2- We are not particularly impressed with transposon screens either, however, if the authors' results, with defined genetic deletion strains, are similar to what has been reported in transposon mutagenesis studies there is no novelty there.

R2. We are NOT aware of any transposon mutagenesis study that has shown the results (or similar) presented in this manuscript. All that was known from those screens is that certain Tn insertions in the *Mtb* genome render the bacilli unable to grow on an unsupplemented agar plate. It was unknown if *thrA*, *thrB* or *metX* mutations are bacteriostatic, bactericidal or just slow down growth, or if they are conditional. It was unknown if such mutations can be complemented, i.e. if the genes are really responsible for the phenotype observed (often Tn mutations have polar effects or other issues). It was unknown to date if *M. tuberculosis* can scavenge methionine, threonine, homoserine from the host during chronic infection. It was unknown if this pathway is essential for persistence. It was unknown so far that *M. tuberculosis* exports lysine. All of these aspects of the aspartate biosynthesis pathway are important to know prior to initiating drug development and to better understand the regulation of amino acid metabolism in *Mtb*. Hence the claim, that the comprehensive work presented here is no more novel than an undirected transposon screen that did not investigate the physiologic *in vitro* or *in vivo* roles of these enzymes specifically, is completely unjustified.

Q3- Finally, we think the authors are over- and mis-interpreting lysine "secretion". The authors have not really demonstrated secretion by *lysE* and certainly not demonstrated lack of secretion by the *lysE* deletion strain. All they showed is that there is more lysine out, which could be derived from dead cells or from another transporter.

Q3.1 Have the authors analyzed other amino acids, unrelated to aspartate/threonine/lysine/arginine/methionine? Is the overall level of metabolites in spent media higher in these mutants?

R3.1.: Thank you for this comment. We have indeed performed these experiments. The figure below shows that arginine and lysine but no other amino acids accumulate. Moreover, the intermediates amino adipate and n-acetyl lysine that accumulate intracellularly by up to 100-fold (Figure 3 in the manuscript), do not leak out of the cells, which indicates that the cells were still intact and lysine/arginine export is a specific process. We have added this figure to the supplementary material (Figure S6) and added a reference to the text (Line 386).

Figure S6. Only lysine and arginine abundance changes in the supernatant of *Mtb ΔthrA*, indicating that the export of these amino acids is specific and not due to general export/leakage of intracellular metabolites. Results are representative of 3 biological replicates of at least 2 independent experiments. *** p-value <0.001 in student t-test.

Q3.2 Also, the title of figure 5 indicates and the figure shows "co-transport". Are the authors sure lysine and arginine are being co-transported (at the same time)? If yes, which experiments in the paper prove that?

R3.2. We agree with the reviewer that co-transport (export at the same time) is not mechanistically demonstrated in this study, however the cumulative evidence points to this with very high certainty:

1. The LysE homolog in *Corynebacterium glutamicum* competitively exports lysine and arginine (doi: 10.1099/00221287-147-7-1765).
2. Only lysine and arginine are enriched in the supernatant in the *thrA* mutant after 2 days of starvation, a condition in which the *lysE* (Rv1986) gene is induced 10-fold.
3. Only Rv1986, and no other LysE homolog, is induced under the conditions where lysine and arginine are accumulating in the supernatant (*thrA* strain).
4. LysG, the regulator of LysE in *Corynebacterium glutamicum*, activates transcription of *lysE* in the presence of lysine or arginine (10.1099/00221287-147-7-1765)
5. Deletion of *lysE* or its regulator *lysG* leads to a 10-fold increase in susceptibility to toxic lysine and arginine (but not histidine) analogs (Figure 5).
6. Rhizobia are resistant to canavanine (toxic arginine analog) due to the presences of a LysE exporter. Deletion of the LysE exporter makes them hypersusceptible to the toxic plant metabolite and leads to inability to infect roots (doi:10.1111/j.1365-2958.2009.06790.x)

Despite this, we have tempered our conclusions and changed the wording to "exported". The arrows in figure 5 are now dashed to indicate that co-transport is only indicated.

Q3.3 Furthermore, the experiments with *lysE* deletion strain are interesting. However, why not show that deletion of the transporter you are claiming is transporting lysine and arginine out, abrogates transport? Without showing that lysine/arginine "secretion" is significantly diminished in this deletion strain, the authors are omitting an important result. This manuscript cannot be published claiming that *lysE* is secreting two amino acids without showing that.

R3.3. We have discussed this extensively in the last round of revisions.

Regarding the assertion that we have not demonstrated the lack of secretion of lysine/arginine in a Δ *lysE* mutant, we respectfully disagree with this conclusion. The toxic amino acid analog export experiments have been a widely accepted method for demonstrating amino acid exporter activity (e.g. 10.1128/JB.02505-14, 10.1111/j.1365-2958.2009.06790.x, 10.1128/JB.186.11, 10.1128/JB.05869-11). We see no reason why this should no longer be the case, and the reviewer has provided no justification for why this standard should be changed.

As to the assertion that we have not "really" demonstrated the secretion by the wild-type strain (and abrogation of it in a *lysE* mutant), the secretion of lysine is only detectable in the *thrA* or *thrB* mutants. We have not yet identified a condition for wildtype *Mtb* where export of lysine is detectable (most likely due to feed-back inhibition by threonine), hence the only way of showing this experimentally is by making a double *thrA/lysE* mutant or by the use of toxic amino acid analogs. As mentioned in the manuscript, we were unable to get a double mutant in multiple attempts, yet we could show 10-fold increase in susceptibility to toxic lysine and arginine analogs. We also attempted to demonstrate the proposed function of LysE as a lysine carrier/exporter protein by a large number of biochemical experiments (performed in the Cook lab). These focused on developing a heterologous expression system for LysE production using *E. coli* and *M. smegmatis* expression strains. The ultimate goal was to obtain pure LysE for reconstitution studies into liposomes for transport studies with [¹⁴C]-L-lysine. We attempted numerous strategies to obtain purified LysE including a wide range of affinity tags, fusion constructs, codon optimized constructs, expression strains, temperatures. However, none of these were successful in obtaining purified LysE. It should be noted that there are no published reports of LysE protein production from any bacterial species for functional studies. We have extensive expertise with membrane protein expression and functional analyses so we are in a very strong position to perform these studies. However, the cumulative evidence from our study and work published by others (see R3.2) allow us to propose that lysine in *Mtb* is exported through Rv1986.

Q3.4 The results presented with amino acid analogs were interpreted by the authors as supporting their statement, however if their hypothesis was correct, and *lysE* was secreting lysine and arginine, upon deletion, no secretion should be observed. Consequently, intracellular lysine and arginine would increase, which would be protective against toxic analogues. This is the exact opposite of what the authors have. Hence, without data on the diminished secretion of lysine and arginine by the *lysE* deletion and the increase in toxicity of these amino acid analogs we can only conclude that *lysE* is not an exporter of these amino acids, as the authors claimed.

R3.4. We respectfully disagree with this view. The accumulation of toxic analogs will by far outweigh the concentration of the canonical amino acids. A *lysE* mutant under standard *in vitro* growth conditions does not accumulate high amounts of lysine, because feedback inhibition by threonine is still intact. Arginine does not accumulate in the *thrA* mutant, because its biosynthesis is not under feedback repression of

threonine. Only in a *thrA/lysE* double mutant would we expect lysine to accumulate to even higher concentrations.

Reviewer #2 (Remarks to the Author):

The revised version of this manuscript is a vastly improved document which has largely addressed the major concerns identified in the first round of reviews; the authors have included a significant amount of additional experimental data which strongly support their central claims. Having considered their thorough rebuttal and accompanying manuscript, only the following substantive concern remains:

Q1. Given the heavy reliance in this study on a set of mutants comprising key gene inactivations and complementations – especially for *in vivo* experiments – it is critical to know whether the authors checked these strains throughout the experimental work (i.e., before and after inoculation/ infection) for the integrity of phthiocerol dimycocerosate (PDIM) biosynthesis (doi:10.1128/JB.00166-10)? This is not only highly relevant to the virulence/pathology phenotypes reported (doi:10.1099/mic.0.029199-0) but, considering the structure and composition of the mycobacterial cell membrane (doi:10.1146/annurev-micro-091014-104121), is almost certain to impact also the respective capacities of these strains for efficient amino acid uptake/export.

R1. We thank the reviewer for this valuable comment. In fact, we performed whole genome sequencing of the parental strain and two clones of the *thrA* mutant and found no differences in PDIM related genes, however we did not check all strains throughout the experimental work, because complementation of the phenotype is shown. Also, we found no significant differences in PDIM gene expression in our transcriptome analysis of $\Delta thrA$ and $\Delta thrB$ compared to WT, which excludes potential regulatory or transcriptional defects. Furthermore, WT and both conditional knockdown strains (without doxy) show almost identical virulence in mice. We are aware of the fact that PDIM mutations have caused bias in virulence studies (e.g. doi: 10.1128/IAI.00097-11), yet these were related to mutant phenotypes that cannot be complemented. Given the clear virulence of the parent strain and the two conditional knockdowns and clear complementation of the phenotypes, we would argue confidently that inactivation of the respective gene is the cause for the observed attenuation, and not potential PDIM mutations. We have also made MetX mutants in the clinical isolate *Mtb* CDC1551, a completely different strain, and we see the same *in vivo* attenuation. The reviewers make an interesting point that loss of PDIM could potentially influence amino acid uptake and export. Consulting the literature though, we could not find a single reference that shows a correlation of PDIM production/loss and amino acid transport (or any other nutrient transport). There is limited evidence that a PDIM mutation increases the permeability of the cell wall to SDS and chenodeoxycholate, however the phenotypes were not complemented (DOI 10.1074/jbc.M100662200). In any case, PDIM mutations would be expected to facilitate rather than inhibit transport across the cell wall. Our results clearly show that threonine, methionine and homoserine can rescue growth *in vitro*, which is proof that these strains can transport those amino acids, but not *in vivo*. Moreover, PDIM deficiency was shown to attenuate growth of *M. tuberculosis* only in mouse lungs but not in other organs and is not required for persistence in the lungs of mice (DOI:10.1038/47042). Our results show that the selected aspartate pathway enzymes are also needed during chronic infection and in the spleen. Based on these observations, we consider it highly unlikely that potential PDIM mutations could account for any of the phenotypes observed in this work.

Q2. I have also been asked to provide further feedback regarding the concerns raised by Reviewer #1. This reviewer has identified three major concerns with this work, in particular that the inter-related lysine "secretion"/ lysine-arginine co-

transport claims are overstated and not adequately supported by the experimental data presented here.

To me, this is the most compelling of the Reviewer #1's arguments and, therefore, a potentially fatal flaw in this work. Consequently, having considered the manuscript and the reviewer's comment in this regard, I support the demand for further experimental evidence for this. The reviewer has made some reasonable and achievable suggestions to this end:

a. Have the authors analysed other amino acids, unrelated to aspartate/threonine/ lysine/arginine/methionine? Is the overall level of metabolites in spent media higher in these mutants?

This is critical, and easily done.

Please see response R3.1 for reviewer 1.

b. Provide at least one piece of experimental evidence proving the claimed co-transport effect.

This is also critical.

Please see response R3.2, R3.3 and R3.4 for reviewer 1.

These additional experiments will add value and, having considered the reviewer's comments, I agree are essential for publication

Reviewer #3 (Remarks to the Author):

I am satisfied with the authors response to my critique.

REVIEWERS' COMMENTS:

Reviewer #2 (Remarks to the Author):

I am satisfied that the authors have largely addressed the pressing concerns highlighted in the previous round of reviews. A substantial amount of experimental work has been added to the manuscript to alleviate all nagging doubts, and the authors should be given credit for this. Clearly, the phenotypes consequent on disruption of the "aspartate pathway" are complex; in fact, if anything, this complexity slightly undermines the authors' claim that the pathway represents a potentially untapped source of novel antimycobacterial compounds. It seems a lot of additional work will be required to demonstrate unequivocally (and to convince TB drug developers) that this target is as attractive as claimed in a more complex (non-human) primate model and/or clinically. That said, the article submitted here is of significant value to the field, mostly in extending the range of known phenotypes associated with disruption of core metabolic pathways.

Minor comments:

1. In their rebuttal letter, the authors report the difficulties encountered in attempting to generate a thrA/lysE double mutant. Did they ever consider using knock-down technology to construct a regulable lysE mutant in the thrA background, or vice versa?
2. Recent work from the Baughn lab (doi: 10.1128/mSystems.00070-19) has highlighted the impact of growth medium on inferred gene essentiality; given the extensive discussion throughout the review/rebuttal process around the value (and limitations) of Tn screens, my preference would be to see some of that correspondence raised in the final manuscript, together with appropriate citations such as that highlighted in this comment.

REVIEWERS' COMMENTS:

Reviewer #2 (Remarks to the Author):

I am satisfied that the authors have largely addressed the pressing concerns highlighted in the previous round of reviews. A substantial amount of experimental work has been added to the manuscript to alleviate all nagging doubts, and the authors should be given credit for this. Clearly, the phenotypes consequent on disruption of the "aspartate pathway" are complex; in fact, if anything, this complexity slightly undermines the authors' claim that the pathway represents a potentially untapped source of novel antimycobacterial compounds. It seems a lot of additional work will be required to demonstrate unequivocally (and to convince TB drug developers) that this target is as attractive as claimed in a more complex (non-human) primate model and/or clinically. That said, the article submitted here is of significant value to the field, mostly in extending the range of known phenotypes associated with disruption of core metabolic pathways.

We thank the reviewer for these positive comments. We have toned down the language about the translational value. However, we respectfully disagree about the additional work required to engage TB drug developers, as our data has already resulted in a collaboration with a major pharmaceutical company to screen against our targets.

Minor comments:

1. In their rebuttal letter, the authors report the difficulties encountered in attempting to generate a thrA/lysE double mutant. Did they ever consider using knock-down technology to construct a regulable lysE mutant in the thrA background, or vice versa?

We have not attempted to make the lysE hypomorph in the thrA background (or vice versa), but this remains an experiment for future work in this area.

2. Recent work from the Baughn lab (doi: 10.1128/mSystems.00070-19) has highlighted the impact of growth medium on inferred gene essentiality; given the extensive discussion throughout the review/rebuttal process around the value (and limitations) of Tn screens, my preference would be to see some of that correspondence raised in the final manuscript, together with appropriate citations such as that highlighted in this comment.

We have added the following sentences (including the mentioned reference) to the discussion: "To date, in vitro transposon sequencing (Tn-seq) screens have been used to predict gene essentiality in Mtb^{43,44} and have guided target selection in drug discovery. The usefulness of Tn-seq screens for the TB research field is undisputed, yet their limitations for drug discovery need consideration. A recent study highlighted the impact of growth medium on inferred gene essentiality⁴⁵ and illustrated how in vitro Tn-seq screens can be limited predictors of in vivo essentiality and cannot differentiate between lethality, bacteriostasis, slow growth phenotypes or conditional essentiality, all information that is crucial for assessing druggability. Since M. tuberculosis is an intracellular pathogen, essentiality and druggability of targets have to be tested during chronic infection in order to prevent false targets being taken forward for drug development."